# An energy-free strategy to elevate anti-icing performance of superhydrophobic materials through interfacial airflow manipulation

Jiawei Jiang [1], Yizhou Shen [1]✉, Yangjiangshan Xu[1], Zhen Wang[1], Jie Tao [1]✉, Senyun Liu[2], Weilan Liu[1,3] & Haifeng Chen [4]

Superhydrophobic surfaces demonstrate excellent anti-icing performance under static conditions. However, they show a marked decrease in icing time under real flight conditions. Here we develop an anti-icing strategy using ubiquitous wind field to improve the anti-icing efficiency of superhydrophobic surfaces during flight. We find that the icing mass on hierarchical superhydrophobic surface with a microstructure angle of 30° is at least 40% lower than that on the conventional superhydrophobic plate, which is attributed to the combined effects of microdroplet flow upwelling induced by interfacial airflow and microdroplet ejection driven by superhydrophobic characteristic. Meanwhile, the disordered arrangement of water molecules induced by the specific 30° angle also raises the energy barriers required for nucleation, resulting in an inhibition of the nucleation process. This strategy of microdroplet movement manipulation induced by interfacial airflow is expected to break through the anti-icing limitation of conventional superhydrophobic materials in service conditions and can further reduce the risk of icing on the aircraft surface.

The unexpected icing phenomena on the aircraft surface is always inevitable when it traverses the supercooled cloud layer, which causes serious energy and security problems[1–3]. Therefore, numerous approaches have been proposed to impede the growth of ice layer in order to eliminate safety risks during flight, such as reducing the contact of supercooled droplets on the surface[4,5], delaying the ice nucleation process[6], inhibiting the expansion of ice crystals and decreasing the adhesion strength between the ice and the surface[7,8]. Depending on the considerations of additional energy consumption and equipment, these methods can be classified into active and passive anti-icing technologies[9].

Conventional active anti-icing technology always has various shortcomings including short-term validity, excessive energy consumption and intricate equipment[10,11]. Even for the widely utilized electrothermal de-icing method, apart from increasing the energy burden of aircraft during flight, it also introduces undesirable electromagnetic interference on the aircraft surface[12]. Consequently, in light of the objective to reduce droplet-surface contact, the superhydrophobic anti-icing technology with droplet self-ejection effect is identified as a promising passive anti-icing approach[13].

Observations have previously demonstrated that various droplet bouncing behaviors on the superhydrophobic surface can be achieved by adjusting the surface structure and wettability[14–16]. Moreover, the contact time between the droplet and the surface can be minified by four times on certain superhydrophobic surfaces[17]. Recently investigations have revealed that numerous air pockets retained on the superhydrophobic surface can diminish the supercooling between droplets and the substrate at low temperatures, resulting in the

[1]State Key Laboratory of Mechanics and Control for Aerospace Structures, Nanjing University of Aeronautics and Astronautics, No. 29 Yudao Street, Nanjing 210016, China. [2]key Laboratory of Icing and Anti/De-icing, China Aerodynamics Research and Development Center, 6 Erhuan South Rd., Mianyang 621000, PR China. [3]Institute of Advanced Materials, Nanjing Tech University, 30 Puzhu South Rd., Nanjing 210009, PR China. [4]Department of Materials Chemistry, Qiuzhen School, Huzhou University, 759# East 2nd Road, Huzhou 313000, PR China. ✉e-mail: shenyizhou@nuaa.edu.cn; taojie@nuaa.edu.cn

postponement of ice nucleation[18,19]. Furthermore, ice nucleation can be delayed by controlling the surface energy and microstructure under low humidity condition[20]. Particularly, the superhydrophobic surface can even prevent droplets from freezing for 7360 s in a static environment of −10 °C[21]. Although the anti-icing performance of superhydrophobic material has been widely recognized, these materials often begin to freeze within seconds under real flight conditions due to the enhanced heat exchange between the microdroplets and the surface which is induced by high-speed impact of numerous subcooled microdroplets and low-temperature airflow[22]. Meanwhile, a certain subcooled microdroplets may be directly pinned and frozen inside the microstructure on the superhydrophobic surface, leading to an anti-icing efficiency limitation of about 30% without the assistance of external field[23–27]. To further enhance the anti-icing property, photo-thermal materials such as cermet, metal oxide and carbon-based materials are introduced into the superhydrophobic surface. Afterwards, solar radiation can be converted into heat to further impede the nucleation process of supercooled droplets on the superhydrophobic surface[28,29].

It is noteworthy that despite the anti-icing efficiency of superhydrophobic surface has been effectively improved through introducing photothermal materials, it is still difficult to ensure the anti-icing property of surface without sufficient sunlight[30–38]. Even though the superhydrophobic surface was constructed by phase change materials (PCMs), it was still unable to guarantee the all-weather anti-icing ability due to the limitation of energy storage capacity[39]. Hence, drawing inspiration from the concept of anti-icing strategies aided by external field, the ubiquitous wind field accompanying aircraft during flight may also contribute to improving the anti-icing efficiency of superhydrophobic surface around the clock. Previous research demonstrated the significant influence of airflow velocity and surface wettability on the dynamics of single droplet motion[40]. The supercooled droplets on the superhydrophobic surface can be separated from the surface rapidly under the drive of airflow. Furthermore, the arrangement and distribution of numerous microdroplets (around 20 μm in flight environment) were determined by the airflow model near the wall and the liquid water content per unit[41].

Presently, additional surface resistance is always raised by conventional superhydrophobic surface without aerodynamics-designed microstructures under flow conditions, leading to an increase in overall energy consumption during flight. However, superhydrophobic microstructures with specific aerodynamic shapes are expected to directionally control the motion behavior of microdroplets on the surface without affecting the surface resistance. Furthermore, the contact process between the supercooled droplet and the surface can be effectively abbreviated by disturbing the near-wall flow field in low-temperature environment, giving rise to improving the anti-icing capability of the superhydrophobic surface.

In this work, a hierarchical structure surface with both aero-dynamic and hydrophobic characteristics is designed and fabricated in order to control the airflow near the surface, and the anti-icing performance is characterized and analyzed in low-temperature and high-velocity inflow environments. Combined with the simulation and anti-icing experiment, it reveals that the aerodynamic microstructures with certain angles can effectively delay ice formation by disrupting the ordering of water molecules and increasing nucleation barriers. Particularly, the microdroplet flow upwelling induced by interfacial air-flow and microdroplet ejection driven by the superhydrophobic characteristics together provide a higher capability to prevent ice from accumulating on the superhydrophobic surface. The implementation of this strategy can effectively improve the anti-icing performance without relying on specific components and morphology of superhydrophobic materials, expanding the application scope of superhydrophobic materials in the anti-icing field.

## Results
### Microstructure design based on drag reduction
In consideration of the actual flight conditions, the drag of micro-structures with various heights is simulated and analyzed in order to avoid additional drag induced by microstructures, as shown in Fig. 1a. It is evident that the microstructure with a height of 50 μm possesses a higher drag reduction rate of 4.23% under a certain angle of 30°. As the height of the microstructure increases gradually, the drag reduction rate decreases simultaneously. There is even an aggrandizement in surface resistance when the height exceeds 90 μm. With the enlargement of the height, the pressure drag strengthens while the viscous resistance diminishes. Hence, the higher total drag is mainly due to the fact that the reduction of viscous resistance cannot compensate for the growth of pressure resistance. Interestingly, the maximum drag reduction rate still appears on sample with the angle of 30° among microstructures with various angles when the height is limited to 50 μm, as presented in Fig. 1b. Considering the pressure distribution analysis [see Supplementary Fig. 1 and Section 1.1.1 in the Supplementary Information], it is obvious that the viscous resistance decreases first and then increases, reaching the minimum value at an angle of 30°, and the pressure drag continues to ascend with the augment of angle.

Subsequently, specific microstructures with angles of 20° (A-20), 30° (A-30) and 40° (A-40) are selected for flow field analysis, as shown in Fig. 1c. It is indicated that a significant amount of low-velocity fluid is retained within the microstructure of A-20, as displayed in the enlarged image surrounded by a white frame. Meanwhile, the boundary layer thickness in the middle of the arrayed microstructure is about 1.61 mm. Although a certain amount of low-velocity fluid remains inside the microstructure, the thickness of boundary layer decreases from 1.83 mm to 1.35 mm when the angle changes from 30° to 40°. This confirms that the subsidence of low-velocity fluid induced by the raised angle effectively elevates the velocity gradient near the wall, resulting in a growth of viscous resistance. Additionally, the velocity streamlines reveal that the proportion of micro-vortices within the microstructure increases with a greater microstructure angle. Based on the "rolling bearing" effect[42], high-speed fluid flows above the microstructure units instead of along the inner side of the units due to the influence of micro-vortex, and a larger micro-vortex can reduce the contact area between the fluid and the microstructure, leading to a lower surface resistance [see Section 1.1.2 in the Supplementary Information and Supplementary Figs. 2, 3].

Noteworthily, the velocity gradient analysis indicates that a reverse velocity gradient induced by the micro-vortex can provide a thrust, thereby synchronously reducing the surface drag, as shown in Fig. 1d. Moreover, it is evident that the reverse velocity gradient is always decreased with the growth of microstructure angle, leading to a reduction of the reverse thrust. Therefore, in combination with the wall shear stress results [see Section 1.1.3 in the Supplementary Information and Supplementary Figs 4, 5], we infer that the reduction of contact area between fluid and microstructure surface dominates the decrease of viscous resistance when the angle is less than 30°, which is induced by the enlargement of the micro-vortex. However, the lower reverse velocity gradient arising by micro-vortex movement makes the viscous resistance ascend continuously when the angle exceeds 30°.

Based on the aforementioned microstructure design, three samples with specific angles are successfully prepared on the aluminum alloy surface by micro-milling method, as depicted in Fig. 1e. The enlarged images reveal that the surface of these as-prepared samples is smooth with expected geometric parameters and devoid of noticeable defects that may affect fluid flow. Afterwards, a wind tunnel test verifies that the drag reduction effect of the samples is slightly lower than the predicted values and the relative differences are generally maintained within 25%, revealing the reliability of the microstructure design, as shown in Fig. 1f.

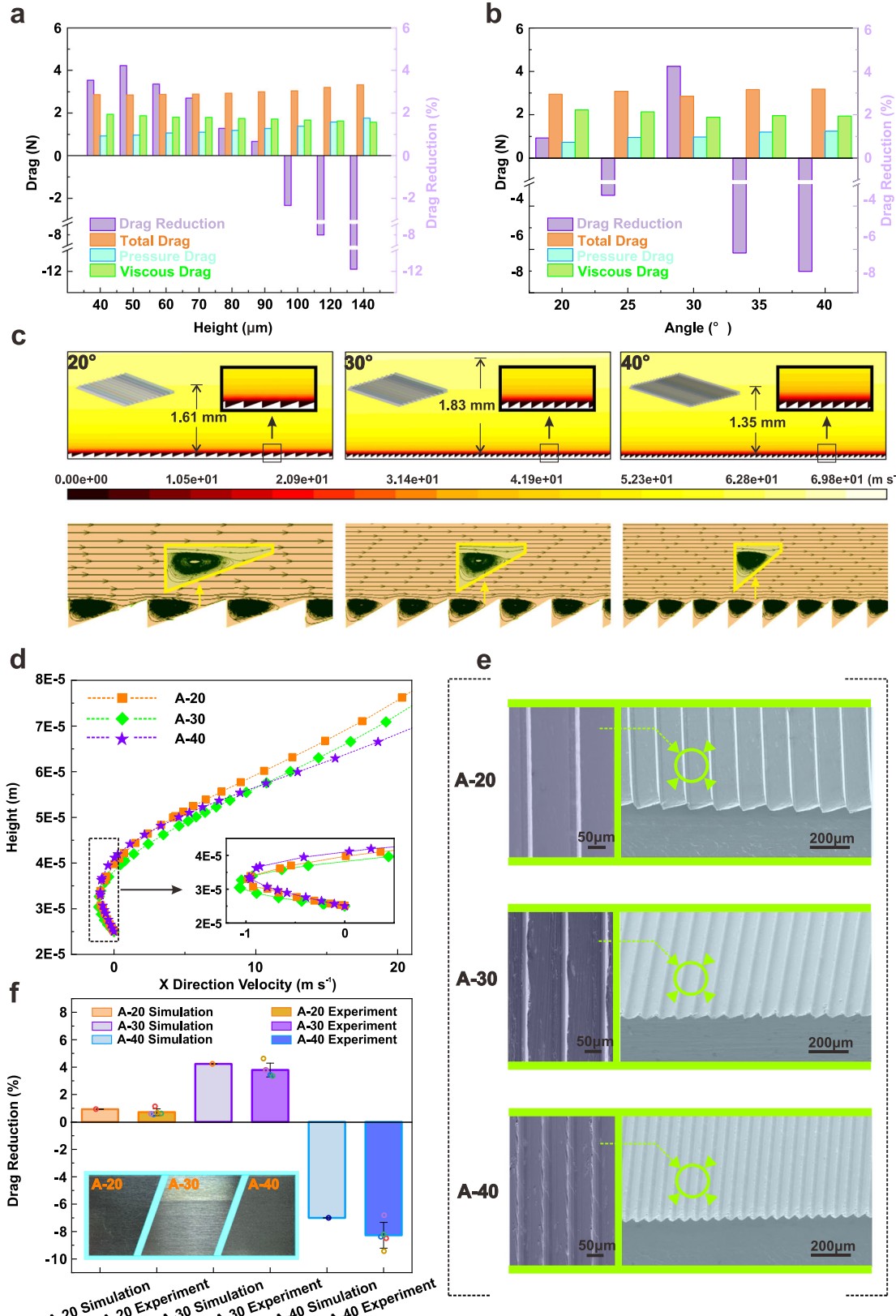

## Superposition of superhydrophobic property on microstructure

Based on the aforementioned considerations, cerium stearate with better hydrophobicity is overlaid on the three samples by an additive manufacturing method of electrodeposition in order to uniformly fabricate superhydrophobic surfaces with both nanostructures and low surface energy without affecting the aerodynamic performance of arrayed microstructures (see Section 1.2 in the Supplementary Information). It demonstrates that all three samples achieve higher water contact angle at 10 min with the augment of electrodeposition time, as illustrated in Fig. 2a. The contact angle (CA) and sliding angle (SA) of A-20 sample are 172.6 ± 0.99° and 1.9° while that of A-30 sample are

**Fig. 1 | Design, fabrication and verification of microstructure under airflow conditions. a** Variation of drag and drag-reduction with height of microstructure. **b** Variation of drag and drag-reduction with angle of microstructure. **c** Velocity distribution analysis of microstructures with angles of 20° (A-20), microstructures with angles of 30° (A-30) and microstructures with angles of 40° (A-40). The white and yellow boxes inserted in this figure show the amplified velocity contour and flow diagram on the microstructure, respectively. **d** Velocity gradient curves with different angles. The reverse velocity gradient is magnified in the dashed frame. **e** Scanning Electron Microscope (SEM) morphology of samples fabricated by micro-milling method. The enlarged view of the typical structure marked by the green circle is illustrated on the left. **f** Drag verification of as-prepared microstructure under wind tunnel environment. The morphologies of these samples are displayed in the orange box, and the small circles in the graphs correspond to raw data. All the samples are tautologically measured for 5 times and averaged to mitigate potential error. Error bars represent standard deviation. Source data are provided as a Source Data file.

170.85 ± 2.13° and 1.7°, respectively. Simultaneously, the A-40 sample exhibits a peak CA of 166.57 ± 2.34° with a SA of 2.1°, which remains higher than that of plate at 10 min. It is inferred that the surface containing arrayed microstructure possesses superior air-capturing capabilities than the simple sedimentary surface. Subsequently, a typical sample of A-20 demonstrates that the surface of the arrayed microstructure is uniform with a few dispersed particles after electrodeposition treatment, as depicted in Fig. 2b. The enlarged image colored by orange reveals that these dispersive particles are spheroidal with a diameter of less than 10 μm and a height of approximately 3 μm, which has little effect on the aerodynamic performance of the arrayed microstructure. Notably, the water droplets are inclined to slide towards the windward side rather than in the opposite direction due to the unique arrayed microstructure, demonstrating an anisotropy in sliding on the superhydrophobic surface [see Section 1.2.1 in the Supplementary Information and Supplementary Fig. 6]. Moreover, the corresponding drag test also verifies this inference [see Section 1.2.2 in the Supplementary Information and Supplementary Fig. 7].

Further amplification analysis of the superhydrophobic surface shows that the deposited surface is mainly composed of a large number of petal-like microstructures about 1.2 μm, which is constituted by a series of interleaved nanorods with a width of around 50 nm. The surfaces of A-30 and A-40 samples also have similar micro-nanostructures due to the identical electrodeposition process [see Section 1.2.3 in the Supplementary Information and Supplementary Fig. 8]. Atomic Force Microscope (AFM) observation on the surface indicates that the root-mean-square roughness (Rqs) of the larger scanning area (10 μm × 10 μm and 20 μm × 20 μm) are 546 nm and 572 nm respectively, while the smaller region shows a lower Rq of 20.2 nm [see Fig. 2c]. It is evidenced that the higher fraction of air can be preserved through vast disordered voids which are introduced by petal-like micro-nanostructures, leading to a better superhydrophobic properties.

Moreover, several chemical analysis methods are used to determine the specific composition of the surface. It can be considered that the absorption peaks at 2850.39 cm$^{-1}$ and 2915.97 cm$^{-1}$ are initiated by the stretching vibrations of methylene (-CH$_2$) group, as revealed by the Fourier Transform Infrared (FTIR) analysis in Fig. 2d and ref. 3. Meanwhile, the new absorption peaks reflect around 1542.83 cm$^{-1}$ and 1452.19 cm$^{-1}$ can be identified as cerium stearate[43].

Further X-Ray Photoelectron Spectroscopy (XPS) analysis reveals that energy spectrum peaks near 284.27 eV and 287.88 eV derived from the C*1s* spectrum are deemed to be −CH$_2$ and O=C−O− respectively, as shown in Fig. 2e. Likewise, peaks around 530.96 eV and 529.18 eV in the O*1s* spectrum can be attributed to the presence of C=O and C−O− bonds[43]. Also, the peaks located at 885.08 eV and 903.08 eV in Ce*3d* spectrum demonstrate the fact that the cerium presented on the surface is considered to be Ce$^{3+}$[44]. The corresponding surface element content measurement shows that the atomic ratio of C, O and Ce is 59.1:6.6:1, as depicted in Fig. 2f. Considering the proportion of related elements in [CH$_3$(CH$_2$)$_{16}$COO]$_3$Ce is 54:6:1, it can be speculated that the [CH$_3$(CH$_2$)$_{16}$COO]$_3$Ce is successfully electrodeposited on the surface. Subsequently, Grazing Incidence X-ray Diffraction (GIXRD) analysis indicates that relevant cerium oxides have not been detected on the surface, which further verifies that the surface composition of the sample is [CH$_3$(CH$_2$)$_{16}$COO]$_3$Ce after electrodeposition treatment[45].

## Evaluation of static icing behavior

The droplet icing test is adopted to investigate the static icing behavior of the droplet on the surface with a hierarchic structure at low temperatures. Regularly, the droplet can be suspended above the structure surface due to its superhydrophobic characteristics when the droplet size exceeds the dimensions of the arrayed microstructure. Figure 3a–d demonstrate the icing behavior of the larger droplets (a diameter of about 2.23 mm) on the surfaces with different hierarchic structures at 253.15 K. It is clear that A-30 sample starts to ice up after 1381 s with a recalescence phenomenon and freezes completely within 19 s, revealing a higher icing-delay performance. In comparison, the icing delay times of A-20 sample and A-40 sample are only 40.2% and 29.7% of that of A-30 sample, respectively. It is clear that the surface structure modification has a significant influence on icing delay although the reduction of surface energy can increase the demand for nuclear energy provided by external systems [see Section 1.3.1 in the Supplementary Information, Supplementary Fig. 9 and Supplementary Fig. 10]. Considering the geometric characteristics of the arrayed microstructure on the superhydrophobic surface, the microstructure with a larger angle can retain more micro-air-pockets [Supplementary Fig. 11]. The interface thermal resistance induced by air-pockets makes it difficult for the latent heat released by preferentially-nucleated ice towards the substrate[46,47]. The latent heat is largely retained inside the droplet, delaying the process of the ice nucleation and growth[48]. Meanwhile, the latent heat release induced by the nucleation of water molecules at the solid-liquid interface may hinder the growth of ice at the interface, resulting in a dry zone between ice and water[49]. This can also reduce the heat transfer efficiency between the droplets and the substrate, effectively delaying the icing process on superhydrophobic surface. Interestingly, the plate treated by the same electrodeposition process has an icing delay time of 505 s, which is even slightly higher than that of A-40 sample. It can be concluded that the wetting state of all the hierarchical structure surface changes from Cassie to Cassie-Wenzel transition state due to the decrease of pressure under low-temperature conditions [see Section 1.3.3 in the Supplementary Information and Supplementary Fig. 12][50]. Consequently, the delayed icing effect is weakened by a vast of solid-liquid interfaces in A-40 sample. It is worth noting that the A-30 sample still shows better anti-icing performance even though these three structural samples basically lose the icing delay property when the temperature is further decreased to 233.15 K [see Section 1.3.3 in the Supplementary Information and Supplementary Fig. 13].

However, the microdroplets (~20 μm in supercooled cloud) are inclined to be embedded and frozen within the arrayed microstructure instead of suspended above the structure. Therefore, the ice accumulation on different samples is adopted to survey the anti-icing performance at 253.15 K since it is difficult to directly observe the liquidus movement behavior of microdroplets, as illustrated in Fig. 3e–h. It is observed that the A-30 sample without hydrophobic treatment achieves a higher ice accumulation than A-20 and A-40 samples at 253.15 K, nevertheless, the A-20 sample has a similar icing phenomenon as the plate. Generally, samples with structure always easier to induce a vast of microdroplets to freeze due to their larger specific surface area. Considering the higher quantity of microstructures in the A-40 sample, it can be speculated that the angle of microstructure has a significant impact on the icing behavior within

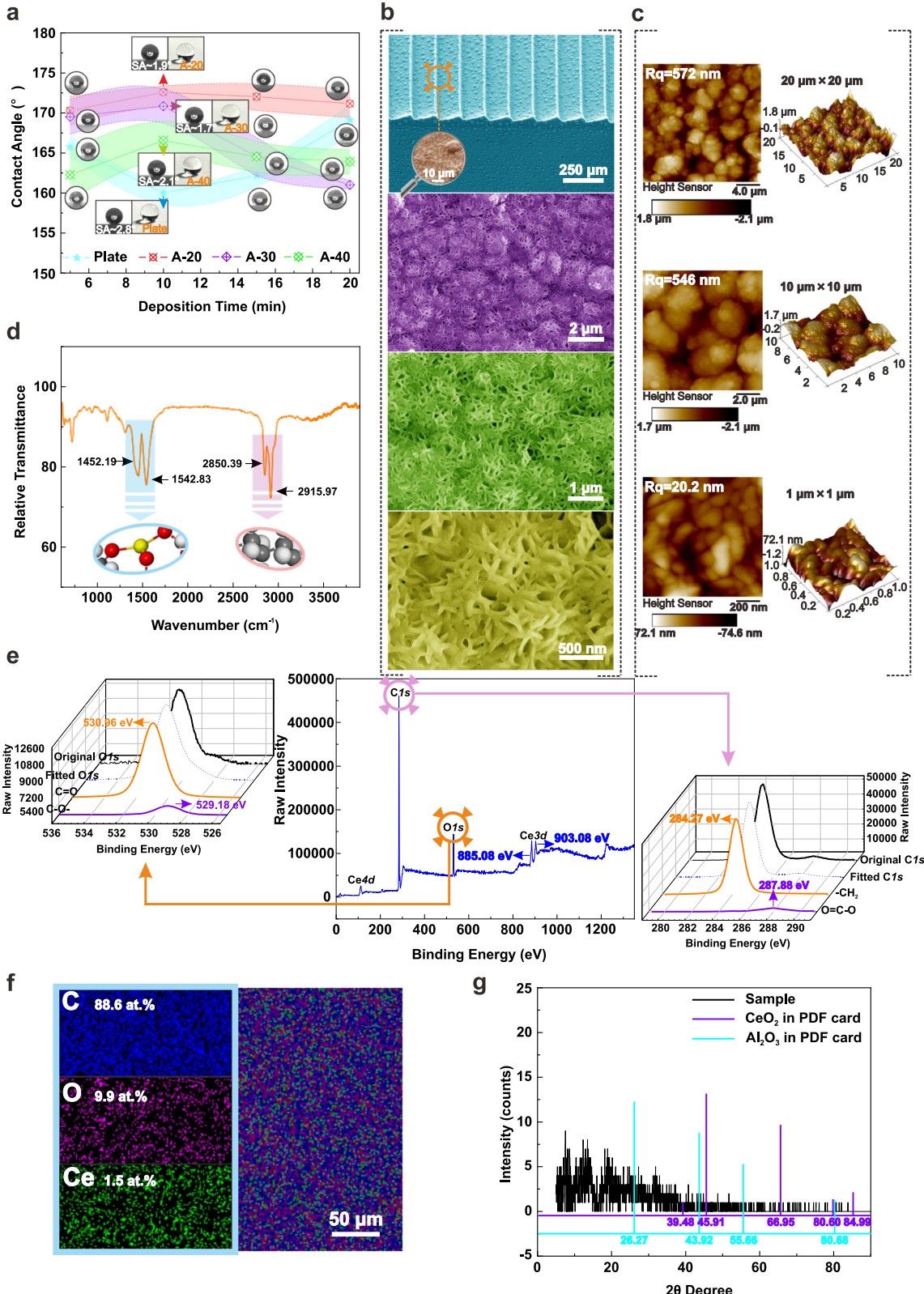

**Fig. 2 | Wettability evaluation and chemical analysis of hierarchical super-hydrophobic surfaces. a** Effects of electrodeposition time on wettability. The actual morphology and contact angle images of droplets on the surface are inserted in this diagram. For all surfaces, five repetitions of measurement are performed at independent locations with a water droplet around 4 μL, and the shaded regions represent the error bands of standard deviation. **b** The SEM morphology of superhydrophobic surface with a microstructure angle of 20° (A-20). The region in the orange circle is selected for observation under different resolutions. **c** The AFM analysis of superhydrophobic surfaces with different resolutions. **d** Surface chemical composition analysis by FTIR. The structure diagrams of the methylene group and the cerium stearate are exhibited within blue and pink ellipses respectively. **e** XPS analysis of as-electrodeposited surfaces. The details of peaks O1 and C1 are indicated as orange and pink arrows respectively. **f** Energy Dispersive Spectrometer (EDS) analysis of as-electrodeposited surfaces. **g** GIXRD analysis of as-electrodeposited surfaces. Source data are provided as a Source Data file.

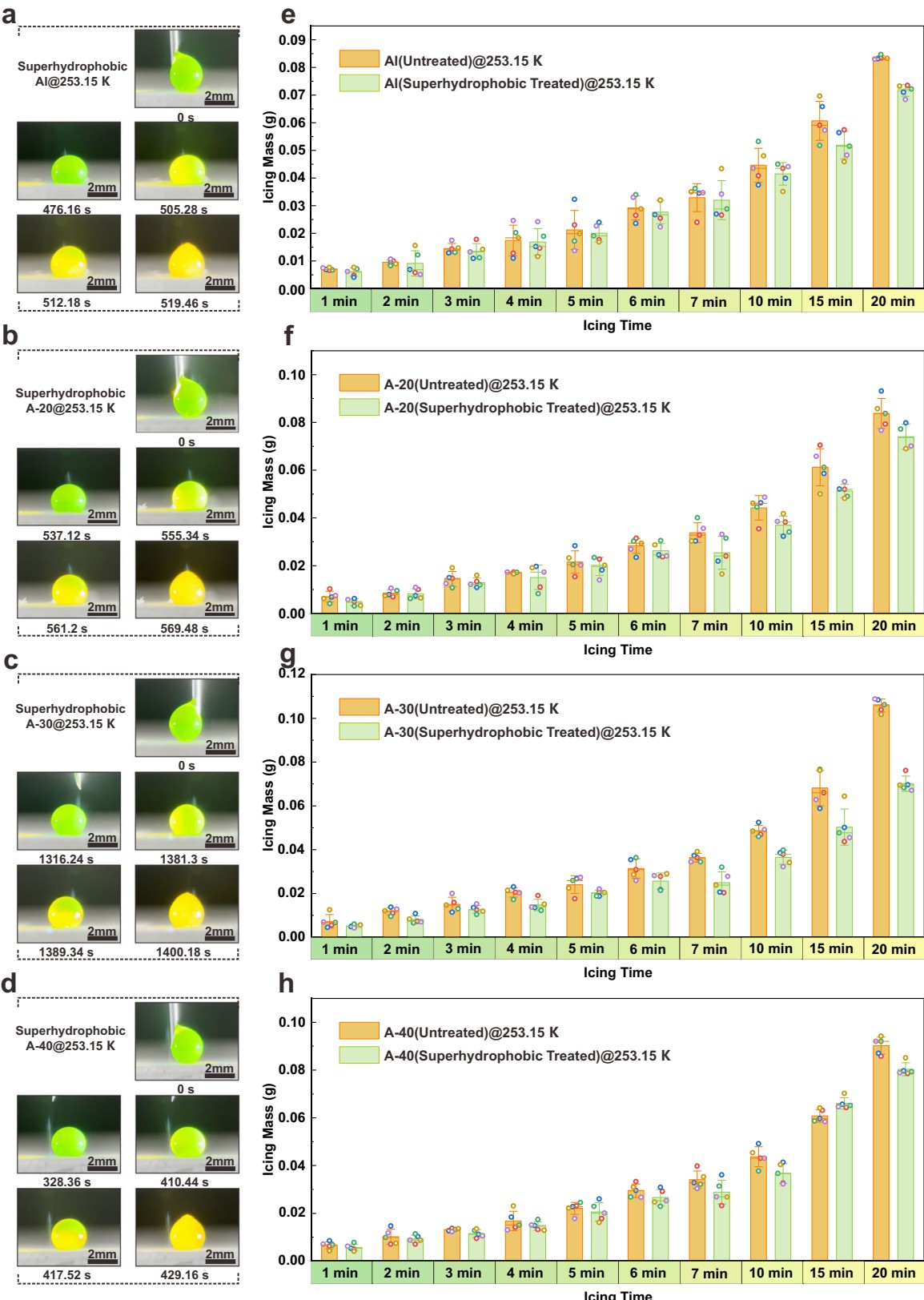

**Fig. 3 | Evaluation of icing delay behavior: Droplet (diameter ~ 2.23 mm) freezes on. a** Al plate at 253.15 K. **b** microstructures sample with angles of 20° (A-20) at 253.15 K. **c** microstructures sample with angles of 30° (A-30) at 253.15 K. **d** microstructures sample with angles of 40° (A-40) at 253.15 K. Microdroplets (diameter~20 μm) freeze on (**e**) Al plate. All the samples are tautologically measured for 5 times and the error bars represent standard deviation. **f** A-20 sample at

253.15 K. All the samples are tautologically measured for 5 times and the error bars represent standard deviation. **g** A-30 sample at 253.15 K. All the samples are tautologically measured for 5 times and the error bars represent standard deviation. **h** A-40 sample at 253.15 K. All the samples are tautologically measured for 5 times and the error bars represent standard deviation. All the small boxes in the graphs correspond to raw data. Source data are provided as a Source Data file.

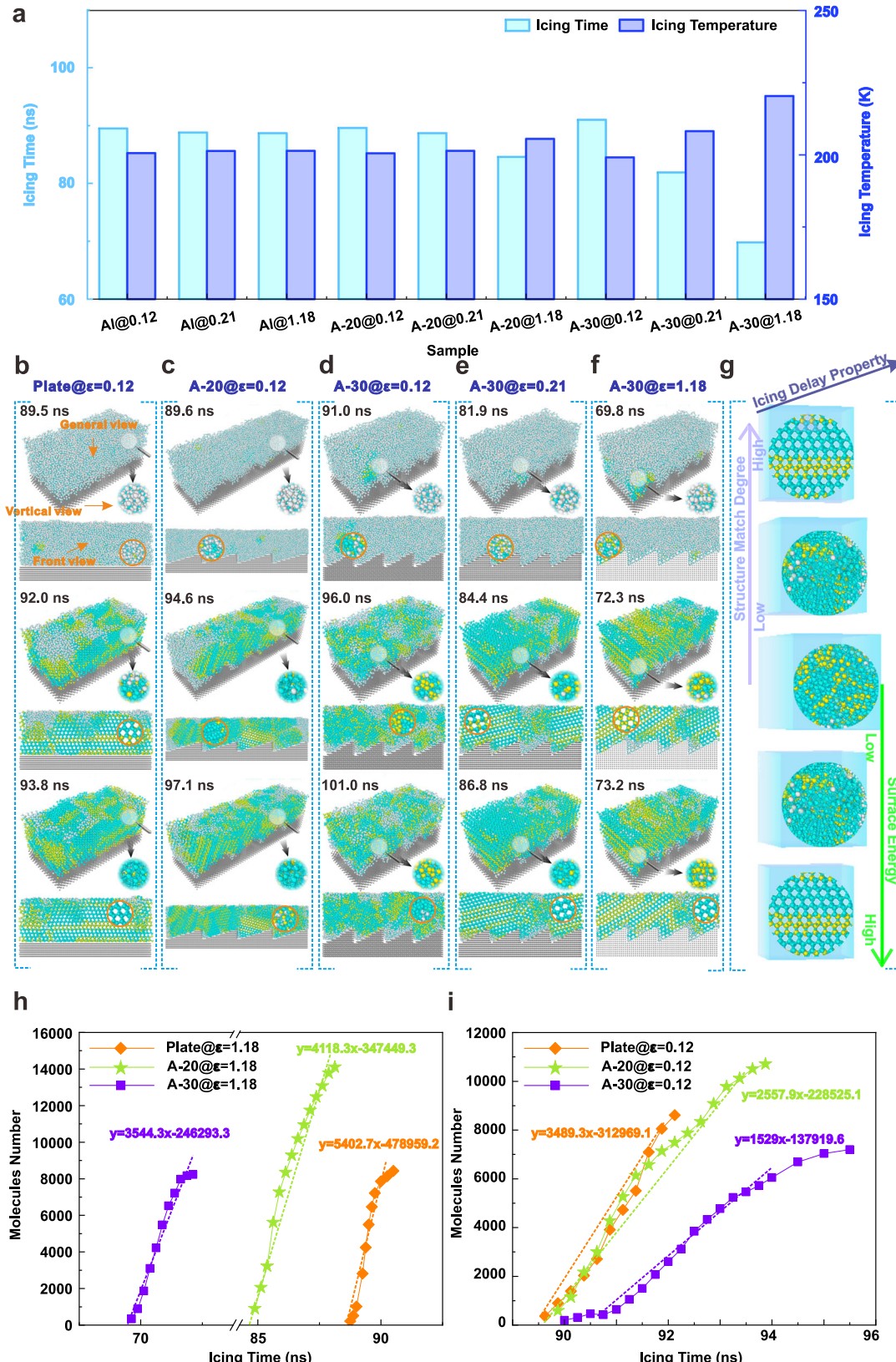

the microstructure[51]. After superhydrophobic treatment, the A-30 sample with less ice accumulation is even superior to the superhydrophobic plate, reflecting a slight advantage in mitigating ice formation induced by microdroplets. It is noteworthy that the superhydrophobic A-30 sample can still maintain lower ice accumulation on the surface at 243.15 K and 233.15 K, respectively [Supplementary Figs. 14, 15].

## Analysis of static icing process of microdroplet

In order to accurately analyze the influence of microstructure angle on the icing behavior, the molecular dynamics method is employed to reveal the static icing process of microdroplets on three representative samples with anti-icing properties (plate, A-20, and A-30), as depicted in Fig. 4. Simultaneously, different interaction energies (0.12 kcal mol$^{-1}$ ~ surface with contact angle of 160°, 0.21 kcal mol$^{-1}$ ~ graphene surface,

**Fig. 4 | Static icing process calculated by molecular dynamics method.**
**a** Nucleation temperature and time. **b** Icing process on the plate with surface interaction energy (ε) of 0.12 kcal mol⁻¹. The arrangement of typical water molecules along the vertical view and front view is amplified within circular regions. The evolution of the initial nucleation area (marked by a magnifying glass) in the general view is enlarged in the circle regions on the right, as indicated by the black arrow. **c** Icing process on microstructures surface with angles of 20° (A-20) at ε of 0.12 kcal mol⁻¹. **d** Icing process on microstructures surface with angles of 30° (A-30) at ε of 0.12 kcal mol⁻¹. **e** Icing process on A-30 with ε of 0.21 kcal mol⁻¹. **f** Icing

process on A-30 with ε of 1.18 kcal mol⁻¹. **g** Underlying mechanism of icing process on the structure surface. The purple and green arrows point to the direction of the structure matching degree and surface energy gradually increasing, respectively, and the arrows with 'Icing delay property' indicate the improvement direction of icing delay performance (higher steps mean better icing delay performance). **h** Ice nucleation rate analyzed under the ε of 1.18 kcal mol⁻¹. The relationship between the number of freezing water molecules and freezing time is illustrated as a dashed line by linear fitting. **i** Ice nucleation rate analyzed under the ε of 0.12 kcal mol⁻¹. Source data are provided as a Source Data file.

and 1.18 kcal mol⁻¹ - aluminum surface) between water molecules and substrate are considered to survey the influence of hydrophobicity on icing process. It is evident that the A-30 model achieves a shorter nucleation time of 69.8 ns with the corresponding nucleation temperature of 220.2 K under the interaction energies of 1.18 kcal mol⁻¹, as illustrated in Fig. 4a and Supplementary Fig. 16. Moreover, the nucleation occurs on the surface of A-20 model after 84.6 ns with an icing temperature of 205.4 K. In contrast, water molecule begins to freeze at 88.7 ns on the plate model, which is consistent with the trend of previous experiment results. When the surface interaction energy is decreased to 0.21 kcal mol⁻¹, the nucleation time of A-30 model is suddenly extended by 12.1 ns to 81.9 ns, indicating that lower surface energy can effectively delay the ice nucleation process. Although the nucleation time of the A-20 and plate model also extends by 5.5 ns and 0.5 ns correspondingly, it can be inferred that the surface interaction energy has a slight impact on these two models. However, the nucleation time of A-30 model is significantly raised to 91 ns when the surface interaction energy is further reduced to 0.12 kcal mol⁻¹, which is even surpassing that of the plate under the same condition. Meanwhile, the nucleation time of A-20 and plate model only increases by 2.52% and 0.11% respectively. This indicates that the specific structural angle can further amplify the de-icing effect.

The detailed nucleation process reveals that water molecules are arranged to hexagonal structure (marked by yellow) preferentially near the plate with a surface interaction energy of 0.12 kcal mol⁻¹, owing to the lower nuclear barrier, as shown in Fig. 4b. Subsequently, cubic crystal emerges near the initial nucleation area (marked by blue) at 92 ns, and a considerable number of molecules near the nucleation site are rearranged into a regular array structure in the front view, as observed in the magnification image marked by orange. This regular structure with a consistent orientation gradually extends throughout the entire ice layer as the icing time progresses [Supplementary Movie 1], some water molecules are also arranged regularly at a certain angle with the normal of substrate. This indicates that certain water molecules can evade the induction of surrounding molecules and grow orderly in multiple directions although the preferential orientation of water molecules exists on the plate. Similarly, the water molecules near the substrate still give priority to form hexagonal ice when the model is altered to A-20, as depicted in Fig. 4c. The hexagonal ice region with regular array expands continuously as the freezing time prolongates, and the nucleation sites appear at various locations above the structure, leading to the formation of mixed regions of hexagon and cube ice, as observed from the front view. Likewise, the regular array of water molecules also exists on the A-20 model perpendicular to the front direction, revealing a slight influence of this structure angle on ice nucleation process under a lower surface interaction energy.

Upon transitioning to the A-30 model, the initial nucleation process resembles that of the previous model, nevertheless, molecules structure with the same orientation are not observed from any perspective with the cooling of the surface [see Fig. 4d and Supplementary Movie 2]. The surface that induces identical orientation of molecules possesses the ability to promote nucleation since the freezing process is essentially a process of regular arrangement of water molecules. It can be assumed that the multi-orientation icing behavior induced by the disruption of ice growth inertia promotes the ice crystals to

constantly adjust their orientation during nucleation in a confined space, exhibiting a postponement of ice nucleation. Therefore, combined with ice type analysis [see Section 1.4 in the Supplementary Information and Supplementary Fig. 17], it can be speculated that the mismatch between ice and substrate induced by the specific angle of 30° further suppresses the nucleation and growth of hexagonal ice, resulting in an inhibition of the nucleation behavior due to the increase of cubic ices with higher energy barriers (cubicity is 0.65 in angle region while 0.6 on plate surface)[42]. Subsequently, the regular hexagonal ice and cubic ice appeared on the A-30 model after 84.4 ns under a higher surface interaction energy of 0.21 kcal mol⁻¹, as displayed in Fig. 4e. Thereafter, several water molecules are only distributed regularly along the front view direction, while partial molecules remain irregular as the freezing time extends to 86.8 ns. Furthermore, it is verified that a significant proportion of water molecules are regularly arranged along the front view direction with the continuous extension of freezing time when the surface interaction energy further increases to 1.18 kcal mol⁻¹ [Fig. 4f and Supplementary Fig. 18]. Therefore, it can be deduced that the enhancement of surface interaction energy can promote the directional arrangement of water molecules along the front view direction on the A-30 structure. The underlying mechanism of icing process on the structured surface is demonstrated in Fig. 4g and the corresponding icing process can be seen in Supplementary Movie 3.

Additionally, according to Fig. 4h, the plate achieves a higher ice nucleation rate when the surface interaction energy is 1.18 kcal mol⁻¹, whereas the nucleation rate of water molecules on the A-30 surface is slower, accounting for 65.6% of that on the plate. This implies that the certain angle of A-30 model can inhibit the subsequent ice growth. In particular, the A-30 model still maintains a lower icing nucleation rate which is only 43.8% of that of plate, when the surface interaction energy is reduced to 0.12 kcal mol⁻¹, as shown in Fig. 4i. These results infer that the existence of a specific structure with low surface energy significantly increases the amount of cubic ice, elevating the icing nucleation barrier, and the probability of multiple nucleation during growth is reduced. Meanwhile, the ice type and growth orientation are constrained to constantly adjust in confined space during the growth of confined space. Under the combination of these two mechanisms, the subsequence ice growth can be inhibited.

## Prediction of dynamic motion behavior for microdroplet

Considering the limited actionability of directly tracking the movement of mass microdroplets, the Fluent Icing method is adopted to predict the movement of microdroplets and the icing behavior of microstructure surface under the practical service conditions. A typical A-30 sample is selected to compare the icing process with the plate. As observed in Fig. 5a, the heat flux at the front of the plate is significantly lower than that at the midsection, while the terminal plate possesses a larger heat flux. This indicates that the significant heat transfer is attributed to the more microdroplets coming into contact with the rear surface when they pass through the plate. In contrast, the heat flux on the A-30 sample exhibits a reverse trend which the heat exchange gradually decreases towards the terminus, as illustrated in Fig. 5b. Particularly, numerous areas inside the microstructure (highlighted by a dashed circle) have an inverted heat transfer. According to

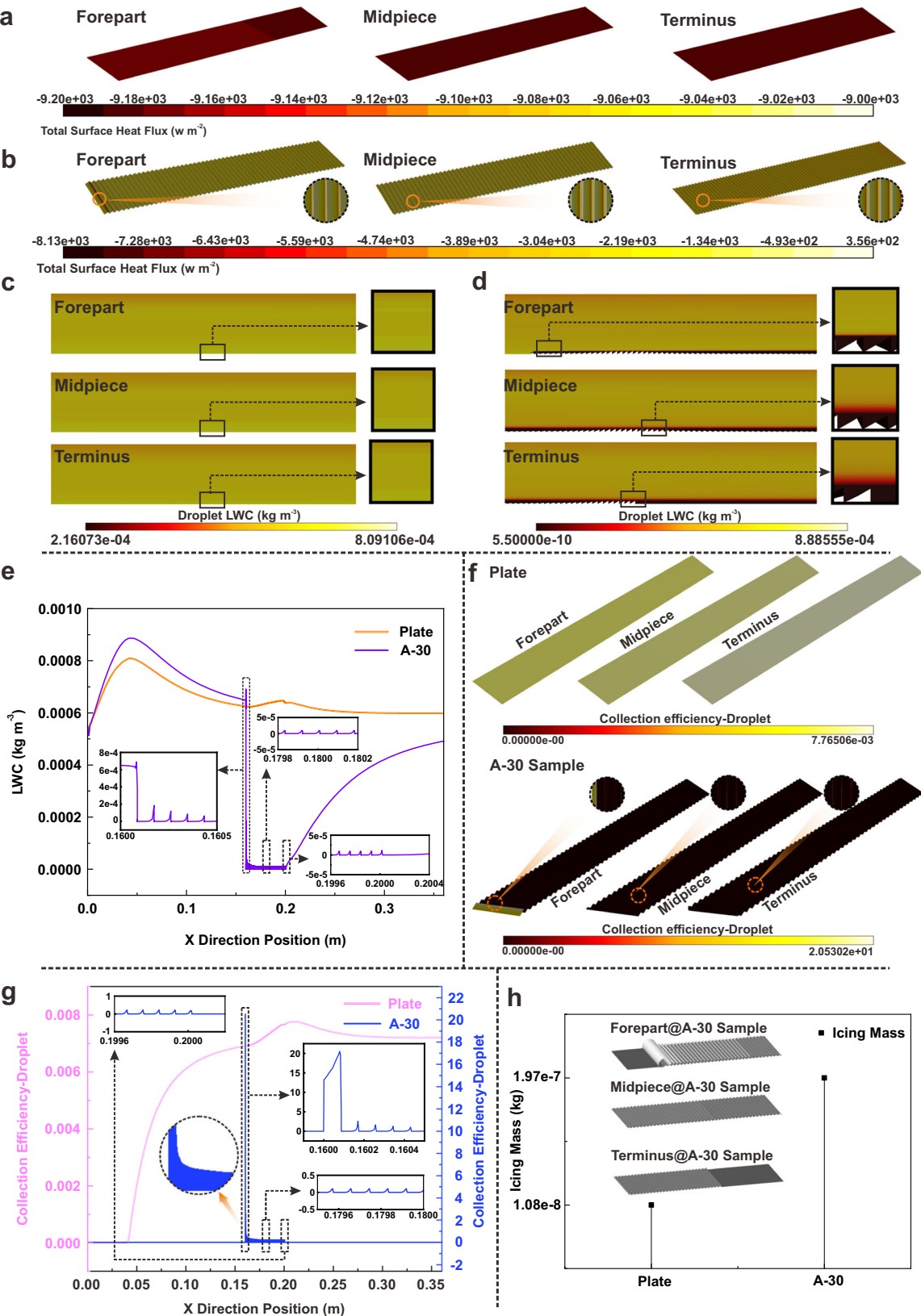

the aforementioned aerodynamic analysis, it can be speculated that the reversed heat gradient is mainly attributed to the friction between the micro-vortex rotating inside the microstructure, and few micro-droplets can enter the microstructure to exchange heat with the substrate.

Moreover, liquid water content (LWC) analysis shows that a wider region of high-density liquid water appears above the front of the plate, as displayed in Fig. 5c. Subsequently, the range of high LWC decreases continuously along the flow direction, revealing a tendency for microdroplets to concentrate towards the surface. Nevertheless,

**Fig. 5 | The CFD simulation of microdroplets movement under flow conditions.** **a** Heat flux of the plate. **b** Heat flux of microstructures surface with angles of 30° (A-30). The heat flux inside the microstructure is amplified and exhibited in the circular regions on the right. **c** Liquid water content (LWC) image of the plate. The distribution of LWC in the near-wall regions of the plate is enlarged and shown in the box on the right. **d** LWC image of A-30. The distribution of LWC in the near-wall regions of the microstructure is magnified and shown in the box on the right. **e** Corresponding LWC curves of plate and A-30. The LWC curves at specific locations (forepart, midpiece, and terminus of the structure region) are amplified respectively. **f** Droplet collection coefficient of the plate (upper section) and A-30 (lower section). The droplet collection coefficient on the microstructure is amplified and displayed in the dashed circular regions. **g** Droplet collection coefficient curves and icing mass of plate and A-30. The curves of droplet collection coefficient at specific locations (forepart, midpiece, and terminus of the structure region) are amplified respectively. particularly, the variation of the droplet collection coefficient on the forepart of the microstructures is magnified into the dashed circle, as indicated by the orange arrow. **h** The icing mass of plate and A-30 samples within 60 s. The Icing morphologies of representative areas (forepart, midpiece, and terminus of the structure region) on A-30 surface are inserted into this figure. Source data are provided as a Source Data file.

the range of low LWC near the substrate of A-30 sample progressively rises along the flow direction, as shown in Fig. 5d. Furthermore, except for a small amount of liquid water remaining in the foremost microstructures, the LWC within the subsequent microstructures remains at a low level, which also confirms the analysis of heat flux. The LWC data extracted from the test area demonstrates that the LWC above the plate is consistently higher than that above the microstructure, as depicted in Fig. 5e. The amplification region in Fig. 5e confirms a sharp decrease in LWC above the microstructure with the increase of microstructure quantity. Based on the above velocity field analysis, it can be considered that the low-velocity fluid can be raised by microstructures, leading to a movement restriction of high-velocity fluid with microdroplets. Considering the hysteresis of microdroplet movement in low-velocity fluid, the existence of the microstructure significantly reduces the distribution of microdroplets above the sample.

The analysis of droplet collection coefficient verifies that the terminal of the plate contacts more microdroplets than the forepart, as observed in Fig. 5f. On the contrary, microdroplets are always attracted to the front of the microstructure sample. It is clear that the forefront microstructure achieves a higher droplet collection coefficient, which may cause severe ice accumulation subsequently. Surprisingly, the microdroplets tend to gather predominantly at the top of the subsequent microstructure with a decreased tendency when the microstructures gradually increase in the flow direction. This observation is also supported by the corresponding droplet collection coefficient curve in Fig. 5g. The ice distribution results confirm that the ice accumulation on the microstructure surface is primarily concentrated at the front of the sample, and even several microstructures at the terminal are almost not covered by ice, as shown in Fig. 5h. The relevant calculation of A-20 sample also reveals a similar conclusion [see the Section 1.5 of the Supplementary Information and Supplementary Fig. 19]. It is disheartening that the ice accumulated on the microstructure sample without superhydrophobic treatment is still more than that accumulated on the plate at the same measurement region [Fig. 5h and Supplementary Fig. 20]. This demonstrates that the reduction of the microdroplet contact caused by the subsequent microstructure is still insufficient to compensate for the excessive microdroplet nucleation at the forefront microstructure. However, the unique icing behavior of the microstructure surface shows advantages of anti-icing performance on a large scale, since the ice accumulation on the plate continues to increase significantly with the length. Therefore, it can be deduced that the surface treated by a superhydrophobic method which can prevent excessive droplet accumulation at the front of the sample may serve as an effective approach to establish the superiority of anti-icing property induced by the microstructure surface.

### Icing behavior of microdroplets in icing wind tunnel

The results of the icing wind tunnel test reveal that the ice on the aluminum plate gradually increases along the flow direction, whereas the ice on the microstructure (A-30) surface exhibits a decreasing tendency, as shown in Fig. 6a. This also confirms the conclusion drawn from previous icing prediction calculations. Additionally, although ice

accumulation on the superhydrophobic plate continues to increase along the flow direction, the icing mass decreases by 20.9% compared with that of the untreated plate at 300 s. This suggests that the superhydrophobic materials can effectively prevent micro-droplets from remaining on the surface and freezing. However, the icing mass on the treated plate increases significantly to 0.0153 g when the icing time is extended to 600 s, surpassing the icing mass on the untreated plate by 11.7%. Surprisingly, compared to the conventional superhydrophobic plate, the A-30 surface with superhydrophobic treatment achieves a lower ice accumulation, which is always 40% less than that accumulated on the superhydrophobic plate. This implies that the coupling of array microstructure optimized by aerodynamics can effectively enhance the anti-icing property of superhydrophobic materials under a high-speed environment. The quality and morphology of ice accumulation on A-20 sample with superhydrophobic treatment also verify this conclusion [see Section 1.6.1 in the Supplementary Information and Supplementary Fig. 21].

Meanwhile, the movement of microdroplets on various superhydrophobic-treated surfaces in a refrigeration tunnel is observed by a high-speed camera, as shown in Fig. 6b–e. The room temperature is also adopted to investigate the influence of temperature variation on microdroplet motion behavior. It is evident that a noticeable bouncing phenomenon of microdroplets appears on the plate at room temperature, which is similar to the traditional impact behavior of larger droplets[46]. However, the microdroplet can only roll on the plate for a brief period at 253.15 K, and struggle to regain its upward momentum [Fig. 6c and Supplementary Movie 4]. This indicates that microdroplets are difficult to detach from the plate at a lower temperature, resulting in an increased risk of icing. Surprisingly, even at a temperature as low as 253.15 K, the bouncing behavior of microdroplets on A-30 sample can still be observed [Fig. 6e and Supplementary Movie 5].

Notably, the motion of the microdroplet is dominated by airflow and is weakly affected by its own inertia and gravity since the size of microdroplet is small. Based on the fact that the plate makes it difficult to maintain the dynamic superhydrophobic property at 253.15 K, it is inferred that the decrease of temperature always leads to the reduction of air pressure inside the superhydrophobic structures, resulting in the decline of the superhydrophobic property. According to the previous simulation of microdroplet motion, it reveals that the low-velocity fluid always converges to the substrate while the velocity gradient augments along the flow direction, leading to a concentration of microdroplets towards the surface. The velocity of microdroplets is significantly reduced as they approach the substrate, and the driving force of the microdroplet ejection is also gradually diminished. Hence, the air-pocket within the superhydrophobic structures can only support the rolling of the microdroplet at 253.15 K. Although there are analogous low-velocity regions above the A-30 sample, the rotation of micro-vortices within the microstructure imparts an additional driving force that facilitates the bouncing of the microdroplet off the surface. This conclusion can also be confirmed by the similar phenomenon observed on the A-20 sample [Supplementary Fig. 22], and the corresponding mechanism diagram is demonstrated in Fig. 6h. It is worth noting that although the A-30 sample has a slight increase of 2.62% at a

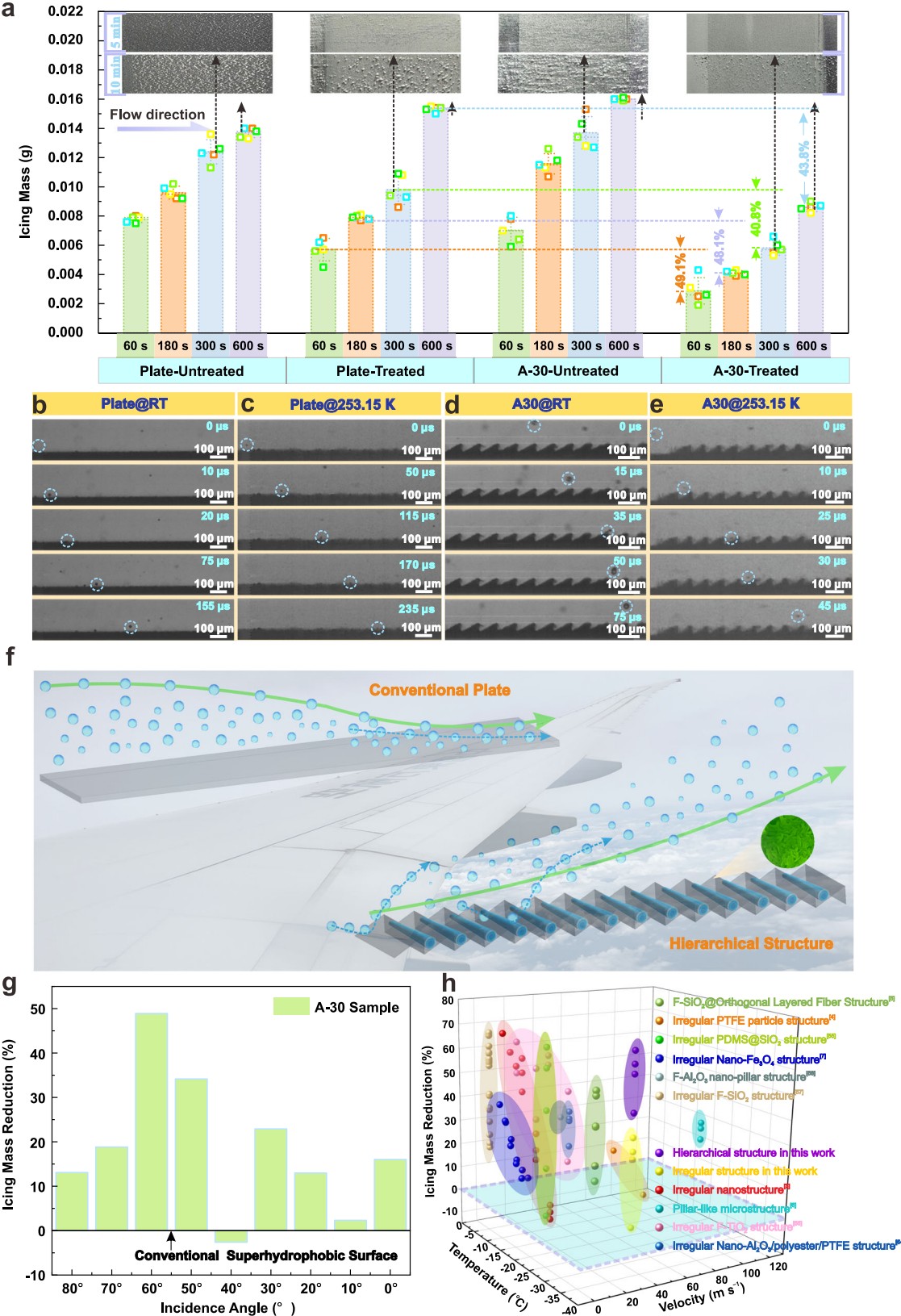

flow incidence angle of 40°, the ice accumulation on the A-30 sample is always lower than that on the conventional superhydrophobic surface at various incidence angles (even at an incidence angle of 0°, the ice accumulation reduction on the A-30 sample is still 16.04%), indicating the widespread applicability of the array hierarchical structure in low-temperature and high-velocity inflow environments [see Fig. 6g, the

Section 1.6.2 of the Supplementary Information and Supplementary Figs. 23 and 24].

In general, the probability of icing on the superhydrophobic-treated hierarchical surface can be reduced through a combination of microdroplet upwelling induced by microstructures and micro-droplet ejection driven by superhydrophobic micro-nanostructures.

**Fig. 6 | Icing behavior of microdroplet in the high-speed icing wind tunnel.**
**a** The morphology and mass of ice. The icing morphologies on the surfaces with the icing time of 300 s and 600 s are demonstrated in the upper part of the graphic respectively. The difference in ice accumulation between the superhydrophobic A-30 sample and the plate under various icing times is represented by corresponding colored numbers (60 s - 49.1%, 180 s - 48.1%, 300 s - 40.8% and 600 s - 43.8%). The small boxes in the graphs correspond to raw data. All the samples are tautologically measured for 5 times and the error bars represent standard deviation. **b** Microdroplets movement on the plate at room temperature. The moving microdroplets are marked by blue dotted circles. **c** Microdroplets movement on the plate at 253.15 K. **d** Microdroplets movement on the microstructure surface with angles of 30° (A-30) at room temperature. **e** Microdroplets movement on the A-30 at 253.15 K. **f** The diagram of microdroplets movement on different superhydrophobic surfaces. **g** Icing mass reduction of A-30 sample. **h** Anti-icing properties of different superhydrophobic materials under various inflow environments[3–7,54–58]. Source data are provided as a Source Data file.

Moreover, the low surface energy also effectively inhibits the icing tendency of structural surfaces with more contact interfaces. Particularly, the microstructure with a specific angle can slightly delay the icing process, and provide more time for the microdroplets to detach from the surface. The implementation of this strategy can further improve the anti-icing performance of the conventional superhydrophobic surface with disordered micro-nanostructure, and even achieve a higher ice accumulation reduction of 62.8% under low temperature and high-velocity flow conditions, as depicted in Fig. 6h.

## Discussion

On account of the actual flight environment, the microstructures were designed and verified in order to minimize additional energy consumption induced by airflow. Afterwards, the superhydrophobic hierarchical structures were successfully constructed on the surface by electrodeposition without affecting the aerodynamic performance. The icing delay behavior revealed that the superhydrophobic hierarchical structure sample with an angle of 30° exhibited a remarkable icing delay time of 1381 s for larger droplets at 253.15 K, and consistently possessed a better icing delay property at lower temperatures. Moreover, the specific angle of 30°contributed to a slight reduction in microdroplet icing on the superhydrophobic surface. This was due to the fact that the disordered arrangement of water molecules induced by the specific angle imposed higher energy barriers for nucleation, thereby inhibiting the icing behavior.

Moreover, the fluid dynamics simulation predicted that the microdroplets were always concentrated above a few microstructures at the front of the microstructure sample and gradually diminished along the flow direction. The subsequent icing wind tunnel test verified that the A-30 sample exhibited superior anti-icing performance, which was only 60% of the icing mass observed on the superhydrophobic plate. Notably, it was difficult for microdroplets to bounce off the plate surface since their movement velocity was greatly decelerated by the low-speed airflow near the wall. Conversely, the microdroplet flow upwelling induced by microstructures and microdroplet ejection driven by superhydrophobic micro-nanostructures together provided a higher capability to reduce the icing likelihood on the superhydrophobic-treated hierarchical surface. The implementation of this strategy may break through the environmental limitations of the current superhydrophobic anti-icing technologies in a certain range, thereby further enhancing the anti-icing efficacy of superhydrophobic on a larger time and space scale.

## Methods

### Design, fabrication, and verification of microstructure

Considering the influence of the microstructure on surface resistance, a drag-reduction microstructure (wedge-shaped structure inspired by barchan dune) was designed and optimized by Ansys Fluent 2022 software[52]. Based on the velocity of aircraft flown through the supercooled cloud layer, the flow velocity was defined as 69.4 m s$^{-1}$. Moreover, the height of the microstructure was set as 40 μm, 50 μm, 60 μm, 80 μm, 100 μm, 120 μm and 140 μm respectively for the sake of disturbing the microdroplet near the wall. Meanwhile, as another typical influence factor, the angle of microstructure was defined as 20°, 25°, 30°, 35° and 40°, respectively. Additionally, the simulation model, detailed calculation parameters and reliability verification were presented in Section 2.1 of the Supplementary Information and Supplementary Figs. 25–28.

On the basis of the optimized microstructure model, a sequence of continuous microstructure arrays was fabricated on the aluminum plate with a size of 30 mm × 30 mm × 2 mm using a five-axis precision milling facility. Subsequently, the surface resistance of these samples was measured by a resistance sensor within a wind tunnel [Supplementary Fig. 29].

### Preparation and characterization of superhydrophobic surface

Based on our prior work[53], a material system of cerium nitrate and stearic with better hydrophobic properties was selected to prepare nanostructure on the microstructure surface by electrodeposition method [see Section 2.2 of the Supplementary Information and Supplementary Fig. 30]. Afterwards, the microtopography of as-prepared samples were observed by SEM and AFM. Additionally, the chemical component and phase were analyzed by FTIR, EDS, GIXRD, and XPS, respectively. The wettability of the superhydrophobic surface was characterized by a contact angle analyzer.

### Evaluation of icing delay behavior

The static freezing process of a single deionized water droplet around 4 μL was captured by a CCD camera. The relative humidity of the test chamber was controlled within 5%, and the reference droplets were positioned on the surfaces of the samples which were cooled to −10 °C, −20 °C, −30 °C and −40 °C respectively. The ice delay time was defined as the duration between the initial contact of the water droplet with the sample and the complete freezing of the droplet.

Numerous microdroplet freezing tests were conducted in a self-made icing environment system including a microdroplet generator and a constant temperature and humidity test chamber [see Section 2.3 of the Supplementary Information and Supplementary Fig. 31]. The size of microdroplet was set to around 20 μm and the freezing temperature was defined as −10 °C, −20 °C, −30 °C and −40 °C similarly.

### Molecular dynamics analysis of static icing process

The molecular dynamics simulation software of LAMMPS (Lammps_3Mar2020) was used to calculate the nucleation process of water molecules on the typical microstructure surfaces. The microstructure is reduced proportionally with a height of 1.5 nm in order to ensure the reliability of modeling and save computing resources. Meanwhile, a water molecular layer with a thickness of 4 nm was incorporated into the simulation to isolate the influence of layer thickness on nucleation icing. Furthermore, the interaction energies between water molecules and substrate were varied to 0.12 kcal mol$^{-1}$, 0.21 kcal mol$^{-1}$ and 1.18 kcal mol$^{-1}$, respectively, so as to assess the influence of surface hydrophobicity on icing behavior [see Section 2.4 of the Supplementary Information, Supplementary Table 1 and Supplementary Figs. 32, 33].

### Icing behavior of microdroplets under low temperature

The distribution of the microdroplets and the icing process were analyzed utilizing the icing module within Ansys fluent 2022 software. Therein, the temperature of the inlet and wall were defined as 253.15 K and 255.54 K, respectively, while the liquid water content was set as

$0.00055\,kg\,m^{-3}$. The corresponding calculation principle, mesh and independence verification are shown in Section 2.5 of the Supplementary Information and Supplementary Figs. 34, 35.

The freezing behavior of the microdroplet was observed by a Charge Coupled Device (CCD) camera in a refrigeration tunnel at $-20\,°C$ to simulate the actual flight environment, and the anti-icing performance was evaluated by calculating the icing quality of different samples. Therein, the velocity of the airflow was controlled at $69.4\,m\,s^{-1}$, and the size of the microdroplet was set around $20\,μm$. Subsequently, the dynamic non-wettability property was captured by a high-speed camera (Photron Mini 100) with a frame of $200,000\,s^{-1}$ at $-20\,°C$. In contrast, the microdroplet motion behavior was also measured at $-10\,°C$ and room temperature.

## Data availability
The data that supports the findings of the study are included in the main text and supplementary information files The Source data that support the findings of this study are provided with this paper. Source data are provided with this paper.

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

## Acknowledgements

The authors acknowledge financial support from the National Natural Science Foundation of China (No. 52075246, received by Y.S.), Natural Science Foundation of Jiangsu Province (No. BK20211568 received by Y.S.), Fundamental Research Funds for the Central Universities (No. NE2022005 received by Y.S.), Basic Research Project of Suzhou (SJC2022032 received by Y.S.), Liaoning Provincial Key Laboratory of Aircraft Ice Protection (XFX20220301 received by Y.S.), Project Funded by the Priority Academic Program Development of Jiangsu Higher Education Institutions (KYCX19_0181 received by J.J.). Project Funded by 2022 large instruments and equipment test fee of Nanjing University of Aeronautics and Astronautics (received by Y.S.).

## Author contributions

J.J. and Y.S. conceived the research. J.J. carried out the overall experiments, flow field simulations and icing process by Fluent. Y.X. and Z.W. performed the icing behavior simulation using molecular dynamics method. S.L., H.C. and W.L. analyzed the experimental data. J.J. wrote the original manuscript and all authors helped revise it. Y.S. and J.T. supervised the research.

## Competing interests

The authors declare no competing interests.
