## [Peer Review File · Nature Communications]

An energy-free strategy to elevate anti-icing performance of superhydrophobic materials through interfacial airflow manipulationEditorial Note: Parts of this Peer Review File have been redacted as indicated to remove third-party material where no permission to publish could be obtained.

REVIEWER COMMENTS

Reviewer #1 (Remarks to the Author):

This study has addressed an important and scientifically significant problem of improving the anti-icing performance of superhydrophobic materials in external environments. By applying a combination of physical and chemical surface modification strategies, it has been found that the ice accumulation on such surfaces can be effectively reduced, provided that the microstructure parameters are appropriate. Although this approach has proven to be effective in achieving high-performance anti-icing properties, there are still some issues that need to be addressed.

1. In the introduction section, the authors suggested “its performance limitation has been gradually approached...” What is the degree of anti-icing performance now? Please provide more numerical values and quantitative representation (efficiency?), and then to facilitate the subsequent presentation of the breakthrough in this work. Besides, what are the real limitations of superhydrophobic materials in terms of anti-icing performance? More related literatures should be introduced to readers.
2. In the Methods section, what is the purpose of the MD simulations of the static icing process as the settings of the substrate do not accord with the experiment one, as well as the cooling temperature? Also, the selection of potential parameters between water and substrates may be not proper, because a complete wetting is always not expected in the field of self-cleaning fields. I suggest that a relationship between ϵ_{ws} and contact angles should be presented, and select these with superhydrophobicity for comparison. How to confirm the cooling rate? Is a timestep of 5 fs very larger? What is the position of models in the simulation box, could they be affected by the periodic nearby models? Besides, what is the purpose of using numerical simulation? Also, Ansys software has been used to analyze the movement behaviors of liquid droplets, but the relevant theoretical basis for CFD is absent. Overall, although multi-scale methods are used, it seems that there are some overlaps in impact and icing studies.
3. In Figure 1, could you explain the irregular changes in viscous resistance and pressure drag with respect to angle, e.g. why 30°? How does the enlargement of the micro-vortex affect the reduction of contact area between fluid and microstructure surface?
4. In Figure 3, could you explain how surface energy affects the icing process? Which one is more important for anti-icing process, surface structures modification or surface energy? Could the transition from Cassie to Cassie-Wenzel happen on the A-20 sample? What is the main factor that decreases the icing delay times for A-20?
5. In Figure 4, how to identify the icing behaviors in MD simulations, and what kinds of ice are obtained in the angled regions, any difference with the flat surface? Also, please show more details about how multi-orientation icing behavior delays the nucleation and icing process. The authors said that a mismatch between ice and substrate induced by the specific angle of 30° may induce a higher energy barrier for nucleation, where does this mismatch come from, atomistic level or any others? also, is 30° a sole one? It is also said that the structures can inhibit the subsequent ice growth, what is the underlying mechanism?
6. In Figure 6, it presents some specific data compared with the plate cases to point out the anti-icing achievement by the coupling of array microstructure optimized by aerodynamics and superhydrophobic

properties, so what is the breakthrough compared with other previous studies that also adopt some microstructural modifications?

Reviewer #2 (Remarks to the Author):

Recently, many researchers related with surface engineering interested in anti-/de-icing, so called icephobicity. Such icephobicity is important in various engineering fields, especially in aircraft. This work would contribute such fields and expected to reduce energy for anti-icing in aircraft. Moreover, the method presented and used in this work manipulating airflow on surface is interesting and show some significant results. Thus I believe this manuscript can be accepted after some revisions. Followings are my suggesting and comment for revision.

I can find some explanation using author's previous work and used citation. However, I recommend to briefly explain previous work, so that the reader do not have to find your previous publication. But, you can still use the citation.

The authors claimed that the surfac with nanostructure is superhydrophobic, but no images of droplet at room temperature. Thus, it would be good to compare the wettability and mobility of water droplet at room tempreature and cold environment.

The grammar should be checked throughout the manuscript. For example, temperature cannot be reduce, but be decreased to XX degree C.

Many researches considere that the icing delay time is related with a transfer of latent heat during the water freezing. So, please include some discussion related with the latent heat of freezing and icing delay time.

The air flow manipulating surface with microscale steps is significanlty smaller than water droplet, so please present the effect of such structure on wettability and mobility of water. is the water droplet has anisotropy in sliding on this surface?

During the anti-icing test, the incident angle is too high. In actual application (i.e., aircraft wing), the problem is leading edge, which has 0 degree of incident angle. Thus, more tests with a low incident angle is required to claim anti-icing.

REVIEWER COMMENTS

Reviewer #1 (Remarks to the Author):

This study has addressed an important and scientifically significant problem of improving the anti-icing performance of superhydrophobic materials in external environments. By applying a combination of physical and chemical surface modification strategies, it has been found that the ice accumulation on such surfaces can be effectively reduced, provided that the microstructure parameters are appropriate. Although this approach has proven to be effective in achieving high-performance anti-icing properties, there are still some issues that need to be addressed.

1. In the introduction section, the authors suggested “its performance limitation has been gradually approached...” What is the degree of anti-icing performance now? Please provide more numerical values and quantitative representation (efficiency?), and then to facilitate the subsequent presentation of the breakthrough in this work. Besides, what are the real limitations of superhydrophobic materials in terms of anti-icing performance? More related literatures should be introduced to readers.

Author reply: Thanks for your suggestions. Currently, traditional superhydrophobic materials that rely solely on the Cassie wetting model often achieve excellent anti-icing performance in static environments (the droplets even do not completely freeze on the superhydrophobic surface at -10°C until 7360 s [1]). However, these results are often measured by millimeter-scale droplets in a static environment. The anti-icing performance of the above superhydrophobic materials is usually inadequate in the

service environment of airplane, unmanned aerial vehicle and other aircraft. Particularly, these superhydrophobic surfaces often begin to icing within seconds under inflow condition with low temperature (temperature $< -10^{\circ}\text{C}$, airflow velocity $> 10\text{ m/s}$, droplet size between $20\text{ }\mu\text{m}$ - $40\text{ }\mu\text{m}$), which cannot achieve the effect of avoiding surface icing. This is mainly due to the enhanced heat exchange between the microdroplets and the surface which is induced by the dense high-speed impact of numerous subcooled microdroplets, promoting the ice nucleation process. Meanwhile, the low-temperature airflow further intensifies the heat dissipation during the contact process between microdroplet and surface, making it difficult for microdroplets to bounce off the superhydrophobic surface. Moreover, a certain subcooled microdroplets may be directly pinned and frozen inside the microstructure on the superhydrophobic surface, providing nucleation sites for subsequent ice formation [2].

Recently, researchers have focused on exploring the anti-icing performance of superhydrophobic surfaces in low-temperature inflow environments using icing wind tunnels. Although they have made targeted adjustments and improvements to prevent microdroplets from pinning into the microstructure, such as using nanostructure to replace the conventional micro-nano hierarchical structure, nevertheless, the superhydrophobic surfaces that rely solely on the low energy and water repellency brought by disordered/isotropic structures still do not demonstrate satisfactory anti-icing property. Considering different anti-icing time and MVD parameters, the ice accumulation mass of the superhydrophobic surface is usually less than 30% lower than that of the untreated surface, and even some superhydrophobic surface show an

abnormal increase in ice accumulation mass, as shown in **Table 1** [3-7].

Therefore, several active technologies (such as electric heating anti/de-icing technology, piezoelectric anti/de-icing technology, and ultrasonic anti/de-icing technology) and passive technologies (photothermal anti/de-icing technology) are coupled to reduce surface icing in order to further improve the anti-icing performance of superhydrophobic materials under aircraft service conditions. Although the photothermal anti-icing technology has shown certain advantages due to its energy free characteristic, the time and scope of the lighting conditions significantly limit the application and promotion of this technology. Consequently, considering the energy-saving requirement, we propose an anti-icing strategy that utilizes wind field to improve the anti-icing performance of superhydrophobic surfaces without requiring energy consumption.

This strategy aims to further improve the anti-icing performance of common superhydrophobic surfaces with irregular structures through interface airflow regulation induced by designed microstructures. The method does not rely on the composition and surface structural characteristics of superhydrophobic materials, demonstrating a wider applicability for improving the anti-icing performance of conventional superhydrophobic materials.

Table 1 Ice accumulation on superhydrophobic surface in ice wind tunnel

Velocity (m/s)	Temperature (°C)	Icing mass reduction (%)	references
23	-15	-3	[3]
23	-15	-6	[3]

23	-15	0	[3]
90	-15	-3.14	[4]
10	-35	25.05	[5]
120	-20	28.88	[6]
120	-20	26.42	[6]
120	-20	21.64	[6]
10	-12	14.23	[7]
10	-14	12.23	[7]

The corresponding statement has been added to the Introduction section using yellow background.

2. In the Methods section, what is the purpose of the MD simulations of the static icing process as the settings of the substrate do not accord with the experiment one, as well as the cooling temperature? Also, the selection of potential parameters between water and substrates may be not proper, because a complete wetting is always not expected in the field of self-cleaning fields. I suggest that a relationship between ϵ_{ws} and contact angles should be presented, and select these with super-hydrophobicity for comparison. How to confirm the cooling rate? Is a timestep of 5 fs very larger? What is the position of models in the simulation box, could they be affected by the periodic nearby models? Besides, what is the purpose of using numerical simulation? Also, Ansys software has been used to analyze the movement behaviors of liquid droplets, but the relevant theoretical basis for CFD is absent. Overall, although multi-scale methods are used, it seems that there are some overlaps in impact and icing studies.

Author reply: Thank you so much for your positive comments and careful check. In this work, we aim to survey the icing behavior of microdroplets under high velocity inflow conditions. A large amount of microdroplets that impact the superhydrophobic surface can easily break away from the surface before ice nucleation with the assistance of the wind field. However, due to the disordered distribution and scale difference of incoming microdroplets, partial microdroplets can always enter the microstructures and freeze at the bottom, which is similar to a static icing process. Notably, the microdroplet is possible to detach from the superhydrophobic surface again under airflow disturbance when the microdroplet fails to freeze immediately after entering the structure. Therefore, it is necessary to clarify the influence of microstructure on the static icing behavior of droplets in order to accurately analyze the effect of structural angle on the final anti-icing property.

On this basis, we conducted comparative experiments on the static icing behavior of microdroplets on different superhydrophobic surfaces and realized that the icing processes on various surfaces were different. Previous literature has shown that the angle of microstructure has a significant influence on the static freezing behavior [8]. Although the settings of molecular dynamics simulation and experiment are not strictly the same, MD simulation is not used to verify the icing experiment in this work, but the underlying mechanism of angle on static icing behavior is revealed by both the experiment and MD simulation methods. Moreover, the icing process with gradual decreased temperature is adopted instead of constant cooling temperature selected in the experiment in order to save computing resources on the premise of revealing the

static icing mechanism, since the icing process with constant cooling temperature requires an extremely long calculation time. Furthermore, a new anti-icing mechanism of superhydrophobic surface under high-velocity flow condition with low temperature is proposed by considering the influence of microstructure on static icing and dynamic icing synchronously.

For the hierarchical structure with both drag reduction and superhydrophobic properties in this work, the substrate is aluminum alloy and the surface are covered with nanoscale cerium stearate. The water repellency of superhydrophobic surface is mainly caused by the low surface energy of cerium stearate molecules and the micro-nanostructure formed by the agglomeration of cerium stearate molecules. Although the low surface energy of cerium stearate molecules can be directly reflected in MD simulation by constructing cerium stearate molecular chains, the micro-nanostructure with randomness on the surface is difficult to realize by MD simulation. Therefore, the superhydrophobic properties of the actual superhydrophobic samples are approximated by assigning surface interaction parameters.

Moreover, as you mentioned, a complete wetting is always not expected in the field of self-cleaning fields. We recalculate and verify the contact angles of water droplets under different surface interaction energy conditions, as shown in Fig. 1. In this work, the interaction energy of flat surface is set as 0.12 kcal/mol so that its contact angle is about 160° , which is similar to the actual contact angle of the flat plate after superhydrophobic treatment for 10 min. Hence, the ϵ of 0.12 kcal/mol used to describe the icing behavior on superhydrophobic surface is considered credible in this work.

Figure 1. Contact angles of plate surface under different interaction energy: (a) $\epsilon=0.06$ kcal/mol, (b) $\epsilon=0.12$ kcal/mol, (c) $\epsilon=0.16$ kcal/mol, (d) $\epsilon=0.21$ kcal/mol, (e) $\epsilon=0.32$ kcal/mol, (f) $\epsilon=0.48$ kcal/mol, (g) $\epsilon=1.18$ kcal/mol

Furthermore, the relevant literature shows that the ramps are performed with cooling rates of 5 K/ns, 2 K/ns, and 1 K/ns, and only the latter resulted in crystallization of ice [9]. Hence, the cooling rate of 1 K/ns is adopted in this work. Additionally, the above literature also reveals that the equations of motion of water are integrated with the velocity Verlet algorithm with a time step 5 fs in the case of the systems with an open water/vacuum interface and 10 fs for the bulk systems [9]. Meanwhile, the calculation time step can be extended appropriately due to the employment of coarse granulation potential. On this basis, we believe that a time step of 5 fs is acceptable for this calculation.

The models are located at the bottom of the simulation box, and a vacuum layer with a height of 400 nm is set above the models, which can completely eliminate the boundary effect of the periodic boundary [10], as shown in Fig. 2.

Figure 2. The diagram of model setup

It is worth noting that MD simulation is always unbecoming to calculate the impacting and icing processes of abundant microdroplets under actual high-velocity inflow condition. Moreover, available high-speed camera can only observe a spot of microdroplets within extremely limited vision. Therefore, the CFD finite element method is adopted in order to predict the movement of microdroplets on the structural surface on a large scale.

In the process of solving microdroplets movement through CFD, the air and microdroplets could be solved simultaneously (as a two-phase flow). However, since the micro-droplets volume fraction is very small, the two-phase flow is considered a dilute gas-particle flow and thus the governing equations of air and micro-droplets are solved in a segregated manner. The airflow is solved first, followed by the micro-droplet equations. In this manner, the effect of the air on the micro-droplets is considered.

In this work, SST-K- ω model is adopted to describe the flow field. For icing simulation, the non-dimensional k and ω on a wall can be expressed as [11]:

$$k_{\omega}^+ = \max \left\{ 0, \frac{1}{\sqrt{\beta^*}} \tanh \left[\left(\frac{\ln \frac{h_s^+}{30}}{\ln 10} + 1 - \tanh \frac{h_s^+}{125} \right) \tanh \frac{h_s^+}{125} \right] \right\} \quad (1)$$

$$\omega_w^+ = \frac{300}{h_s^{+2}} \left(\tanh \frac{15}{4h_s^+} \right)^4 + \frac{191}{h_s^+} \left[1 - e^{-\left(\frac{h_s^+}{250} \right)} \right] \quad (2)$$

where $\beta^*=0.09$, y^+ and h_s^+ are defined as:

$$y^+ = \frac{\rho u_{\tau} d_w}{\mu} \quad (3)$$

$$h_s^+ = \frac{\rho u_{\tau} h_s}{\mu} \quad (4)$$

Therefore, all the wall values of k and ω are known:

$$k_w = f_w(u_{\tau}, k_w^+) = k_w^+ u_{\tau}^2 \quad (5)$$

$$\omega_w = f_w(u_{\tau}, \omega_w^+) = \frac{\rho \omega_w^+ u_{\tau}^2}{\mu} \quad (6)$$

Moreover, the flow field is modeled by partial differential equations for the conservation of mass, momentum and energy. The conservation of mass for a compressible flow, for example one where the density of the fluid is not a linear function of both pressure and velocity, can be written as:

The Energy Equation can be expressed as follow:

$$\frac{\partial \rho_a E_a}{\partial t} + \vec{\nabla} \cdot \left(\rho_a \vec{V}_a H_a \right) = \vec{\nabla} \cdot \left(k_a \left(\vec{\nabla} T_a \right) + v_i \tau^{ij} \right) + \rho_a \vec{g} \cdot \vec{V}_a \quad (7)$$

Where E and H are the total internal energy and enthalpy, ρ is the density and V is the velocity vector, the subscript a refers to the air solution. Respectively. γ is the ratio of specific heats which equals 1.4 for air (perfect gas), and k is the thermal conductivity, computed in a similar way to the laminar dynamic viscosity.

$$k = C1 \times \frac{T^{\frac{3}{2}}}{T + 133.7} \quad (8)$$

where T refers to the static air temperature in Kelvin, and where the C1 is equal to 0.00216176 W/(mK^{3/2})

Subsequently, the following equation is adopted to compute the dynamic viscosity.

$$\frac{\mu_{\infty}}{\mu_{ref}} = \left(\frac{T_{\infty}}{T_{ref}} \right)^{3/2} \left(\frac{T_{ref} + 110}{T_{\infty} + 110} \right) \quad (9)$$

where $\mu_{\infty} = 17.9 \times 10^{-6}$ pa·s.

The general Eulerian two-fluid model consists of the Euler or Navier-Stokes equations augmented by the microdroplets continuity and momentum equations:

$$\frac{\partial \alpha}{\partial t} + \vec{\nabla} \cdot (\alpha \vec{V}_d) = 0 \quad (10)$$

$$\frac{\partial (\alpha \vec{V}_d)}{\partial t} + \vec{\nabla} \cdot [\alpha \vec{V}_d \otimes \vec{V}_d] = \frac{C_D \text{Re}_d}{24K} \alpha (\vec{V}_a - \vec{V}_d) + \alpha \left(1 - \frac{\rho_a}{\rho_d} \right) \frac{1}{Fr^2} \quad (11)$$

where the variables α and V_{eda} are mean field values of, respectively, the micro-droplet concentration and velocity. The first term on the right-hand-side of the momentum equation represents the drag acting on micro-droplets of mean diameter d. It is proportional to the relative micro-droplet velocity, its drag coefficient C_D and the droplets Reynolds number:

$$\text{Re}_d = \frac{\rho_a d V_{a,\infty} \left\| \vec{V}_a - \vec{V}_d \right\|}{\mu_a} \quad (12)$$

And an inertial parameter:

$$K = \frac{\rho_a d^2 V_{a,\infty}}{18 L_{\infty} \mu_a} \quad (13)$$

The second term represents buoyancy and gravity forces, and is proportional to

the local Froude number:

$$Fr = \frac{\|V_{a,\infty}\|}{\sqrt{L_\infty g_\infty}} \quad (14)$$

Certainly, the results of liquid water content and droplet collection coefficient have a similar presentation due to the aerodynamic characteristics of the arrayed microstructures. However, the liquid water content mainly reveals the distribution of microdroplet flow above the microstructures, while the droplet collection coefficient focuses on the probability distribution of microdroplets on the surface. Subsequently, the icing behavior of microdroplets on the structural surface is calculated based on the contact state of microdroplet. Therefore, we believe that there is no overlap between microdroplet movement calculation and icing behavior calculation in this work.

The corresponding statement has been added to the manuscript and supporting file using yellow background.

3. In Figure 1, could you explain the irregular changes in viscous resistance and pressure drag with respect to angle, e.g. why 30°? How does the enlargement of the micro-vortex affect the reduction of contact area between fluid and microstructure surface?

Author reply: Thanks for your suggestions. The flow velocity was set as 69.4 m/s and the height of microstructure was defined as 50 μm in order to study the influence of angle on drag reduction. The variations of drag and drag reduction ratio on angle are shown in Fig. 3. As displayed by histogram, it is clear that the drag reduction ratio decreases firstly, and then increase with the improvement of the angle. The peak value

of drag reduction ratio is achieved at 30°, which is corresponding to the angle in nature. With the further growth of the angle, the drag reduction ratio decreases from 4.23% to -7.10% sharply, showing a significant enlargement of drag. Drag analysis reveals that the pressure drag rises from 0.73 N to 1.24 N as the angle. However, the viscous drag decreases from 2.22 N to 1.88N gently firstly, and then improves to 1.94 N with the augment of the angle.

Figure 3. Variation of drag and drag reduction ratio with different angle of microstructure

Figure 4 shows the variation of pressure around a microstructure unit with different angle. Pressure contour map illustrates that small areas with high pressure and low pressure appear on the windward side and slip face of the microstructure respectively when the angle is raised from 20° to 25°. In addition, as the angle increases, the areas of both the high and the low pressure gradually enlarge. Meanwhile, the difference between the high and the low-pressure areas is also augmented, indicating an increment of pressure gradient. As the angle is raised to 30°, the area of high pressure on the windward side significantly expands to the entire microstructure. Although the

pressure gradient remains stable essentially, the pressure difference between the front and rear of the microstructure still augments. When the angle is further reached to 35° , the negative pressure area on the slip face side is reduced slightly. Simultaneously, a region of high pressure over 50 Pa appears on the wind-ward side, which improves the pressure difference significantly. Subsequently, as shown in Fig. 4e, it is indicated that the pressure of the flow field around the wind-ward side is closed to 50 Pa as the angle is up to 40° . Moreover, although the negative pressure area is dropped sharply, a high-pressure region over 100 Pa appears on the windward side, resulting in a higher pressure gradient. The continuous enlargement of the pressure difference at the front and rear of the microstructure brings a continuous improvement of pressure drag which is consistent with the result of the resistance analysis. Furthermore, as the angle rises, the number of structures in the same calculation region increases accordingly, resulting in a higher accumulation of pressure drag.

Figure 4. Pressure contour map of different angle: (a) 20° , (b) 25° , (c) 30° , (d) 35° ,
(e) 40°

Careful consideration is given to characterize the flow field with different angle in order to fully understand the variation of viscous resistance. The corresponding velocity streamlines diagram of angle is presented in Fig. 5. It is obvious that the denseness of

the streamlines in the micro-vortex remains steady, indicating a stable rotation velocity of micro-vortex with the growth of the angle. It is verified that a relatively stable velocity gradient is achieved between the micro-vortex and the upper fluid during the augment of the angle. In addition, the micro-vortex in the microstructure with an angle of 20° occupies about half of the microstructure in the flow direction, as displayed in Fig. 5(a). Nevertheless, when the angle of the microstructure is increased to 40° , the micro-vortex occupies approximately 80% of the microstructural area in the flow direction, as shown in Fig. 5(e). This is suggested that the size of the micro-vortex augments gradually in the flow direction with the augment of angle, leading to a reduction of the contact area between the fluid and the microstructure. As mentioned above, the increscent size of micro-vortex can effectively reduce the positive viscous resistance. Moreover, it is worth noting that the shape of the micro-vortex is greatly affected by the microstructure. The bigger the angle of the microstructure, the easier it is to accumulate the low-speed fluid at the bottom of the microstructure, lifting the micro-vortex. Thereby, the distance between the micro-vortex and the microstructure surface extends with the growth of angle, which probably result in a decline of reverse velocity gradient near the surface.

Figure 5. Velocity streamlines diagram of structures with different angle, (a) 20° , (b)

25°, (c) 30°, (d) 35°, (e) 40°

The units of wall shear stresses with different angle are listed in Fig. 6. It can be found that both the positive and the negative wall shear stress possess a same downward trend with the enlargement of angle. Moreover, the viscous resistance of a microstructure unit obtained by the integral of the wall shear stress curve has a declining tendency identically. According to the velocity distribution analysis above, the influence of the angle with tiny variation on boundary layer thickness should be slightly since the height of the microstructure is limited to 50 μm . Thus, it can be confirmed that the increased size of micro-vortex can effectively decrease the contact area between fluid and microstructure, leading to a reduction of positive viscous resistance. However, the removed micro-vortex dominates a continuous decreasing of reverse viscous resistance, resulting in a reduction of reverse velocity gradient near the surface. The corresponding curve of reverse velocity gradient is shown in the Fig. 7. It is also evidenced that the curve of reverse velocity gradient moves towards to right with the growth of angle, demonstrating a reduction of reverse velocity gradient.

Figure 6. X direction wall shear stress units with different angle

Figure 7. Reverse velocity gradient curves with different angle

Therefore, combined with the results of drag analysis, we infer that the reduction of contact area between fluid and microstructure surface dominates the variation of viscous resistance when the angle is less than 30° , which is induced by the enlargement of the micro-vortex. Nevertheless, the reverse velocity gradient arose by micro-vortex movement has a greater impact on the viscous resistance when the angle is greater than 30° .

Notably, fluid will flow from the top of the microstructure units instead of along the inner wall of the microstructure units due to the presence of micro-vortex [12]. The gas-gas contact between the micro-vortex and external fluid replaces the original gas-solid contact between the fluid and the wall. The micro-vortex is similar to the roller bearing that converts sliding friction (caused by the wall and fluid) into rolling friction, leading to friction reduction, as shown in Fig. 8.

Based on the “rolling bearing” effect, it is suggested that the size of the micro-vortex augments gradually in the flow direction with the augment of angle, leading to a reduction of the contact area between the fluid and the microstructure. As mentioned

above, the increasing size of micro-vortex can effectively reduce the positive viscous resistance.

[REDACTED]

Figure 8. The mechanism of drag variation under high flow velocity condition

Therefore, combined with the results of drag analysis, we infer that the reduction of contact area between fluid and microstructure surface dominates the decrease of viscous resistance when the angle is less than 30° , which is induced by the enlargement of the micro-vortex. However, the lower reverse velocity gradient arose by micro-vortex movement makes the viscous resistance ascend continuously when the angle exceeds 30° .

The corresponding explanation has been added to the supporting file using yellow background.

4. In Figure 3, could you explain how surface energy affects the icing process? Which one is more important for anti-icing process, surface structures modification or surface energy? Could the transition from Cassie to Cassie-Wenzel happen on the A-20 sample? What is the main factor that decreases the icing delay times for A-20?

Author reply: Thanks for your question. During the heterogeneous nucleation process of ice under the same temperature condition, the energy compensation provided by the

substrate for nucleation can be expressed as:

$$W_{hetero}^* = \frac{16\pi\sigma_{IS}^3 v_I^2}{3\Delta\mu_{IS}^2} \varphi(\theta) \quad (15)$$

Where

$$\varphi(\theta) = \frac{(1 - \cos \theta)^2}{4} (1 + \cos \theta) \quad (16)$$

Here W_{hetero}^* indicates the energy compensation for the formation of a nucleus of ice in substrate in case of heterogeneous nucleation, σ_{IS} is the surface energy of the substrate interface, v_I is the molar volume of ice, $\Delta\mu_{IS} = v_I(p_S - p_I)$ is the difference between the chemical potentials of molecules in water molecule and substrate if the thermodynamic state of substrate remains unchanged during nucleation. The $(p_S - p_I)$ is the pressure difference between the two phases, and the coefficient $\varphi(\theta)$ is determined by water contact angle on substrate. It can be considered that the decrease of surface energy can reduce the energy compensation obtained from the substrate, and then increases the demand for nucleation energy provided by the external system, resulting in the delay of the icing process.

Generally, the influence of surface energy on anti-icing performance is limited. We achieved superhydrophobic surfaces with similar micro-nanostructures but different surface energies on the same aluminum plate by controlling the carbon chain length of organic acids ($[\text{CH}_3(\text{CH}_2)_{12}\text{COO}]_3\text{Ce}$ and $[\text{CH}_3(\text{CH}_2)_{14}\text{COO}]_3\text{Ce}$, where the longer the carbon chain, the lower the surface energy). The corresponding surface topography is shown in Fig. 9.

Figure 9. The morphology of super-hydrophobicity surfaces with different surface energies observed through SEM and AFM: (a)~(c) $[\text{CH}_3(\text{CH}_2)_{12}\text{COO}]_3\text{Ce}$, (d)~(f) $[\text{CH}_3(\text{CH}_2)_{14}\text{COO}]_3\text{Ce}$

The anti-icing experiment shows that there is no obvious discrepancy in the icing delay time of superhydrophobic surfaces with different surface energies, as shown in Fig. 10. However, for the superhydrophobic surfaces with different microstructure in this work, even if the surface energy is consistent, the icing delay time of sample A-30 is 800 s-900 s higher than that of other samples. This indicates that the surface structures modification is more important for anti-icing process.

Figure 10. Icing delay time process on different sample surfaces: (a) $[\text{CH}_3(\text{CH}_2)_{12}\text{COO}]_3\text{Ce}@-15^\circ\text{C}$, (b) $[\text{CH}_3(\text{CH}_2)_{14}\text{COO}]_3\text{Ce}@-15^\circ\text{C}$

Moreover, water molecules at the solid-liquid interface inevitably freeze and nucleate with the decrease of temperature. This leads to the transition of the contact interface from water-solid to water-ice. Meanwhile, the pressure of the air-pocket inside the micro-nanostructure of the superhydrophobic surface can also be reduced as the temperature drops, weakening the support effect of the superhydrophobic surface, and causing an enlargement of solid-liquid interface. Therefore, under the coordination of the above two mechanisms, the transition from Cassie to Cassie-Wenzel also happens on the A-20 sample.

In this work, considering the geometric characteristics of the arrayed microstructure on the superhydrophobic surface, the larger the structural angle, the smaller the space occupied by droplets within the microstructure, as shown in Figure 11. The microstructure with a larger angle can retain more micro-air-pockets to hinder temperature transfer, leading to an improvement of anti-icing performance. This is why the anti-icing performance of A-20 sample is lower than that of A-30 sample.

Figure 11. Diagram of wetting behavior of superhydrophobic surfaces with different microstructures

However, for droplets with fixed size, the larger the structural angle, the more solid-liquid contact interfaces within the same contact radius. Excessive structural angle tends to easily cause a wide range of direct temperature transfer from the low-temperature substrate, promoting the nucleation of droplets. Therefore, there was an

abnormal decrease in the icing delay time on A-40 sample.

The corresponding statement has been added to the manuscript and supporting file using yellow background.

5. In Figure 4, how to identify the icing behaviors in MD simulations, and what kinds of ice are obtained in the angled regions, any difference with the flat surface? Also, please show more details about how multi-orientation icing behavior delays the nucleation and icing process. The authors said that a mismatch between ice and substrate induced by the specific angle of 30° may induce a higher energy barrier for nucleation, where does this mismatch come from, atomistic level or any others? also, is 30° a sole one? It is also said that the structures can inhibit the subsequent ice growth, what is the underlying mechanism?

Author reply: Thanks for your question. The free energy of ice is always lower than that of water under supercooled conditions, hence, there has been a sudden drop in system energy when water molecules start to nucleate. The total potential energy data show that there are four stages in the freezing process [13]: (1) a long quiescent period with relatively constant potential energy; (2) a short period during which the potential energy slowly decreases; (3) a short period during which the potential energy decreases rapidly; and (4) a final period with reduced but relatively constant potential energy and during which the ice structure fully forms. Water in the quiescent period is in a supercooled liquid state, exhibiting intermittent collective motions and energy fluctuations associated with hydrogen bond rearrangements. The freezing process starts

in stage (2). The fact that the system explores the overall relatively flat potential energy landscape for a considerable time (that is, the quiescent period) before entering the fast-growing period agrees with the predictions of basic nucleation theory [14-16]. However, the MD simulation also provides a molecular-level illustration of the water freezing process not obtainable from conventional nucleation theory [16]. The corresponding nucleation temperature and time can be achieved from Fig. 12.

Meanwhile, the ice nucleation is the ordering process of disordered water molecules. The icing process can be captured by monitoring the structure of water molecules [17]. Water molecules in ice are usually arranged in a cubic or hexagonal structure, and the ice structure can be identified using a molecular visualization software OVITO. The recognition path of this software is as follows [18]: First, the nearest neighbors of an atom are identified. Then, for each of these four neighbors, their respective nearest neighbors are identified. This yields the list of second nearest neighbors of the central atom. Finally, the CNA fingerprint is computed for these 12 second nearest neighbors and the central atom. If they are arranged on an FCC lattice, then the central atom is classified as cubic diamond. If they form an HCP structure, then the central atom is marked as a hexagonal diamond atom.

Figure 12. The energy curve of icing process

The ice crystal type is mainly divided into cubic ice and hexagonal ice. Hexagonal ice has a lower nucleation barrier and is theoretically prone to preferential formation. However, it is difficult to form a tight structure during the ice growth process due to the anisotropy on the hexagonal ice structure. Therefore, the ice layer is prone to form a mixed structure of cubic ice/hexagonal ice [19]. Generally, the lattice arrangement of

the plate has a certain effect on the ice crystal structure. However, the aluminum lattice used in this work is not inclined to assist in the nucleation of cubic and hexagonal ice [20]. Therefore, the ice that exists on the plate always presents as a mixed cubic/hexagonal form (cubicity of 0.6 with the $\epsilon=0.12$ kcal/mol, as shown in Fig. 13). It is noted that the cube and hexagonal structures alternate in layers among the mixed ice, as shown in Fig. 14. Previous literature reveals that the microstructure on the substrate has a significant effect on the ice nucleation process, leading to a selectivity of ice type. This also verifies that the limitation of nucleation space has a significant impact on the arrangement of water molecules [21].

Figure 13. The amount of cubic ice and hexagonal ice on different models with $\epsilon=0.12$ kcal/mol

Figure 14. Ice formation on a conventional aluminum plate

In this work, the ice crystals at the initial nucleation sites tended to maintain a regular shape while the ice crystals appear distorted at the junction of two ice nucleus. This is due to the limitation of nucleation space on the conventional arrangement of water molecules, and the ice layer has to adjust its growth phase to adapt to the restriction of nucleation space. Although regular ice cannot form at the interface, the ice within partial angled regions can still maintain a certain degree of order (cubicity about 0.59 with the $\epsilon=0.12$ kcal/mol), as shown in Fig. 15.

Figure 15. Ice formation on the A-20 sample

Meanwhile, the specific angle of the microstructure can contribute to a severe phase deflection of ice crystals, resulting in a completely disordered ice crystal (cubicity about 0.65 with the $\epsilon=0.12$ kcal/mol), as depicted in Fig. 16.

Figure 16. Ice formation on the A-30 sample

Additionally, ice tends to nucleate and grow rapidly along easy growth direction,

and the change of growth orientation is to adapt to the limitation of nucleation space. This change will disrupt the original growth inertia of ice, hindering the growth of the ice layer along the easy growth direction, resulting in a delay in nucleation and growth. Notably, almost no regular ice crystal can be observed on A-30 sample, and the cubic ice and hexagonal ice is not arranged in a layered form, demonstrating a chaotic distribution of ice crystal (this confusion is more chaotic than the cubic/hexagonal hybrid structure). This may be due to the spatial limitations of the icing environment that cause the ice crystals to constantly adjust their orientation during nucleation. This adjustment requires longer time for the nucleation and growth of ice crystals, exhibiting an icing behavior with multi-orientation, and delaying the ice process.

Moreover, the limited growth space is mismatched with the ice type structure due to the unique spatial structure and growth orientation of cubic ice and hexagonal ice, as illustrated in Fig. 17. Ice formation is the process in which water molecules fill the space without creating vacancies as much as possible. Due to the mismatch between ice and substrate, cubic ice and hexagonal ice are always forced to mix and adjust their orientation to achieve space filling. It is obvious that hexagonal ice tends to grow laterally rather than at the top due to its anisotropy. This growth orientation often promotes the formation of hexagonal ice with chain structures, making it difficult to form tight structures in confined environments at atomic level [21].

Figure 17. Structure diagram of cubic ice and hexagonal ice

It is worth noting that cubic ice is inclined to form a tight structure in restricted space due to its isotropic of growth orientation, and can easily stuff the space with adaptive forms such as rotation. The spatial mismatch between ice and substrate induced by the specific angle of 30° is obvious, which further suppresses the nucleation and growth of hexagonal ice, resulting in an increase in the number of cubic ice (cubicity is 0.65 in angle region while 0.6 on plate surface). As a consequence, the appearance of a large amount of cubic ice with higher nuclear energy barriers has led to an overall increase in the nuclear energy barrier of ice formation.

In order to investigate whether only the structures with an angle of 30° possess a special icing delay property, we also intend to conduct icing calculations for structures with angles of 10° , 50° , 60° , 70° , and 80° under the same conditions as previous, respectively. Therein, icing calculation is not performed on surface of structures with an angle of 80° due to the extremely narrow gaps between the structures, making it difficult for water molecules to enter the structure (this model is basically similar to a flat plate). The corresponding simulation results reveal that although the nucleation time of water molecules on different structures varies, the ice layers on all models

exhibit a regular lattice arrangement in a specific direction when the water molecules are completely frozen, as shown in Fig. 18. This indicates that the structure with an angle of 30° is still unique among all computational models.

Figure 18. Icing behavior on different structures with the angles of 10° , 50° , 60° and

70°

Considering the above analysis, we infer that the existence of a specific structure (with an angle of 30°) significantly increases the amount of cubic ice, elevating the icing nucleation barrier, and the probability of multiple nucleation during growth is reduced. Meanwhile, the ice type and growth orientation are constrained to constantly adjust in confined space during the growth of confined space. Under the combination of these two mechanisms, the subsequent ice growth can be inhibited.

The corresponding statement has been added to the manuscript and supporting file using yellow background.

6. In Figure 6, it presents some specific data compared with the plate cases to point out the anti-icing achievement by the coupling of array microstructure optimized by aerodynamics and superhydrophobic properties, so what is the breakthrough compared with other previous studies that also adopt some microstructural modifications?

Author reply: Thanks for your question. Certainly, previous researchers have conducted ice wind tunnel validation on the anti-icing ability of superhydrophobic surfaces with microstructural modifications. However, these investigations mainly carry out under the condition of low-velocity inflow ($< 30\text{m/s}$) and the freezing temperature is always limited to 0°C - 10°C . In this case, superhydrophobic materials with microstructural modifications usually exhibit excellent anti-icing properties, and the ice accumulation is reduced by about 50% compared to the surfaces without superhydrophobic treatment. However, the ice accumulation reduction of these superhydrophobic surfaces is always reduced to about 30% when the flow velocity increases to 50 m/s . The ice accumulation reduction further decreases with the augment of flow velocity and the decline of temperature. This is due to the fact that conventional superhydrophobic surfaces are mainly designed for static icing environments, and there is a lack of consideration for the influence of microstructure on surface aerodynamic performance under inflow conditions. In our work, the ice accumulation reduction of superhydrophobic surfaces with irregular microstructure range from 20%-26% at a flow velocity of 69.4 m/s and a temperature of -20°C . However, the anti-icing property of superhydrophobic surfaces indicates a significant improvement (the ice accumulation reduction is above 50%) when the disordered superhydrophobic micro-nanostructures

are combined into microstructures designed by aerodynamic theory. Particularly, the superhydrophobic surface with hierarchical structure can achieve a higher ice accumulation reduction of 62.8% under certain condition. The comparison results of anti-icing properties of superhydrophobic materials under different inflow environments are demonstrated in Figure 19.

The corresponding statement has been added to the manuscript and supporting file using yellow background.

Figure 19. Anti-icing properties of different superhydrophobic materials under various inflow environments

Thanks again for your valuable suggestions and positive evaluation on our research work.

References:

[1] Wen M, Wang L, Zhang M, et al. Antifogging and Icing-Delay Properties of

- Composite Micro- and Nanostructured Surfaces. *ACS Applied Materials & Interfaces*. 6, 3963 (2014).
- [2] Huang X, Tepylo N, Pommier-Budinger V, et al. A survey of icephobic coatings and their potential use in a hybrid coating/active ice protection system for aerospace applications. *Progress in Aerospace Sciences*. 105, 74-97 (2019).
- [3] De Pauw D, Dolatabadi A. Effect of Superhydrophobic Coating on the Anti-Icing and Deicing of an Airfoil. *Journal of Aircraft*. 54, 490-499 (2017).
- [4] Kimura S, Yamagishi Y, Sakabe A, et al. A New Surface Coating for Prevention of Icing on Airfoil//2007 SAE Aircraft and Engine Icing International Conference. (2007).
- [5] Liu X, Chen H, Kou W, et al. Robust anti-icing coatings via enhanced superhydrophobicity on fiberglass cloth. *Cold Regions Science and Technology*. 138, 18-23 (2017).
- [6] Alamri S, Vercillo V, Aguilar-Morales A I, et al. Self-limited ice formation and efficient de-icing on superhydrophobic micro-structured airfoils through direct laser interference patterning. *Advanced Materials Interfaces*. 7, 2001231 (2020).
- [7] Liu Z, Feng F, Li Y, et al. A corncob biochar-based superhydrophobic photothermal coating with micro-nano-porous rough-structure for ice-phobic properties. *Surface and Coatings Technology*. 457, 129299 (2023).
- [8] Bi Y, Cao B, Li T. Enhanced heterogeneous ice nucleation by special surface geometry. *Nature communications*. 8, 15372 (2017).
- [9] Lupi L, Hudait A, Molinero V. Heterogeneous Nucleation of Ice on Carbon Surfaces. *Journal of the American Chemical Society*. 136, 3156-3164 (2014).

- [10] Haji-Akbari A, DeFever R S, Sarupria S, et al. Suppression of sub-surface freezing in free-standing thin films of a coarse-grained model of water. *Physical Chemistry Chemical Physics*. 16, 25916-25927 (2014).
- [11] Aupoix B. Roughness Corrections for the $k-\omega$ Shear Stress Transport Model: Status and Proposals. *Mathematical Notes*. 20, 240 (2015).
- [12] Song X, Zhang M, Lin P. Skin friction reduction characteristics of nonsmooth surfaces inspired by the shapes of barchan dunes. *Mathematical Problems in Engineering*. (2017).
- [13] Matsumoto M, Saito S, Ohmine I. Molecular dynamics simulation of the ice nucleation and growth process leading to water freezing. *Nature*. 416, 409-413 (2002).
- [14] Krämer B, Hübner O, Vortisch H, et al. Homogeneous nucleation rates of supercooled water measured in single levitated microdroplets. *The Journal of Chemical Physics*. 111, 6521-6527 (1999).
- [15] Bartell L S. Nucleation rates in freezing and solid-state transitions. Molecular clusters as model systems. *The Journal of Physical Chemistry*. 99, 1080-1089 (1995).
- [16] Abraham F F, Zettlemoyer A C. *Homogeneous Nucleation Theory*, Academic Press. New York, NY. (1974).
- [17] Li N, Jiang J, Yang M Y, et al. Anti-icing mechanism of combined active ethanol spraying and passive surface wettability. *Applied Thermal Engineering*. 220, 119805 (2023).
- [18] Maras E, Trushin O, Stukowski A, et al. Global transition path search for dislocation formation in Ge on Si (001). *Computer Physics Communications*. 205, 13-

21 (2016).

[19] Haji-Akbari A, Debenedetti P G. Direct calculation of ice homogeneous nucleation rate for a molecular model of water. *Proceedings of the National Academy of Sciences*. 112, 10582-10588 (2015).

[20] Pedevilla P, Fitzner M, Michaelides A. What makes a good descriptor for heterogeneous ice nucleation on OH-patterned surfaces. *Physical Review B*. 96, 115441 (2017).

[21] Xu Y, Shen Y, Tao J, et al. Selective nucleation of ice crystals depending on the inclination angle of nanostructures. *Physical Chemistry Chemical Physics*. 22, 1168-1173 (2020).

[22] Rivero P J, Rodriguez R J, Larumbe S, et al. Evaluation of functionalized coatings for the prevention of ice accretion by using icing wind tunnel tests. *Coatings*. 10, 636 (2020).

[23] Wang F, Tang R, Wang Z, et al. Experimental study on anti-frosting performance of superhydrophobic surface under high humidity conditions. *Applied Thermal Engineering*. 217, 119193 (2022).

[24] Sharifi N, Dolatabadi A, Pugh M, et al. Anti-icing performance and durability of suspension plasma sprayed TiO₂ coatings. *Cold Regions Science and Technology*. 159, 1-12 (2019).

[25] Yin L, Xia Q, Xue J, et al. In situ investigation of ice formation on surfaces with representative wettability. *Applied Surface Science*. 256, 6764-6769 (2010).

[26] Rico V, Mora J, García P, et al. Robust anti-icing superhydrophobic aluminum

alloy surfaces by grafting fluorocarbon molecular chains. *Applied Materials Today*. 21, 100815 (2020).

Reviewer #2 (Remarks to the Author):

Recently, many researchers related with surface engineering interested in anti-/de-icing, so called lipophobicity. Such lipophobicity is important in various engineering fields, especially in aircraft. This work would contribute such fields and expected to reduce energy for anti-icing in aircraft. Moreover, the method presented and used in this work manipulating airflow on surface is interesting and show some significant results. Thus, I believe this manuscript can be accepted after some revisions.

Followings are my suggesting and comment for revision.

1. I can find some explanation using author's previous work and used citation. However, I recommend to briefly explain previous work, so that the reader do not have to find your previous publication. But, you can still use the citation.

Author reply: Thank you so much for your reminder. As you mentioned, using citation may cause difficulties for readers in understanding the text. We are pleased to briefly describe the critical conclusions from previous work in the text. Relevant statements have been added to the manuscript using yellow background.

2. The authors claimed that the surface with nanostructure is superhydrophobic, but no images of droplet at room temperature. Thus, it would be good to compare the wettability and mobility of water droplet at room temperature and cold environment.

Author reply: Thanks for your suggestion. The wettability and mobility of the various

superhydrophobic surface with nanostructure have been examined and compared under different temperature conditions. The water contact angle (WCA), sliding angle (SA) and droplet morphology images are presented in Figure 1. It can be observed that the flat surface with nanostructures exhibits super-hydrophobicity at room temperature (a WCA of 160.62° and SA of 2.8°). Meanwhile, all three hierarchical structure surfaces demonstrate excellent super-hydrophobicity after combining this nanostructure with the array microstructure (the WCA/SA of A-20, A-30 and A-40 samples are $169.74^\circ/1.9^\circ$, $170.82^\circ/1.7^\circ$ and $167.31^\circ/2.1^\circ$, respectively). Subsequently, the CAs of all these superhydrophobic surfaces are decreased to about 145° when the temperature is decreased to 0°C , while the SAs of these surfaces are unable to obtain due to the wetting between the droplets and the substrate with condensate water. Additionally, the CAs of superhydrophobic surfaces further reduce to around 130° when the ambient temperature drops to -10°C . Notably, the superhydrophobic flat surface shows a lower CA of 124.45° at -10°C . However, the variation of CA on hierarchical structure surfaces is not obvious as the temperature further decreases to -20°C , nevertheless, the CA of the flat surface is reduced by 6° . This indicates that simple nanostructures are difficult to retain the air-pockets captured on the surface in low-temperature environments. Moreover, except for the A-30 sample, the droplets freeze on the surfaces within a few seconds when the temperature drops to -30°C . Furthermore, the droplets freeze immediately upon contact with all the superhydrophobic surface when the temperature decreases to -40°C . Generally, the existence of arrayed microstructures can effectively enhance the water repellency of superhydrophobic surfaces in low-temperature environments.

Figure 1. The wettability and mobility of the various superhydrophobic surface with nanostructure under different temperature conditions

3. The grammar should be checked throughout the manuscript. For example, temperature cannot be reduced, but be decreased to XX degree C.

Author reply: Thank you so much for your positive comments and careful check. The grammatical errors mentioned have been corrected and highlighted in yellow background. Regarding the Language issue, we tried our best to modify these language mistakes and carefully checked the full text again. Furthermore, we invited a native speaking professor in our university to refine the English writing. We believe that the quality of English writing is greatly improved in the revised manuscript.

4. Many researches considered that the icing delay time is related with a transfer of latent heat during the water freezing. So, please include some discussion related with the latent heat of freezing and icing delay time.

Author reply: Many Thanks! In this work, the droplet icing on superhydrophobic surface is a cooling process accompanied by heat exchange between droplet and low-temperature substrate. Therefore, the icing behavior is significantly affected by the heat transfer efficiency between the droplets and the substrate. According to the heat transfer formula at the interface [1]:

$$R=L/KA \quad (1)$$

the interface thermal resistance R is directly proportional to the thermal conductivity distance L , and inversely proportional to the contact area A and thermal conductivity K [1].

Compared with the hydrophilic surface, the solid-liquid contact area of the superhydrophobic surface is smaller due to the presence of air-pockets, significantly increasing the thermal resistance at the solid-liquid interface. It is worth noting that the greater interfacial thermal resistance not only reduces the heat conduction efficiency directly through the solid-liquid interface, but also makes it difficult for the latent heat released by preferentially-nucleated ice towards the substrate [2-5]. Hence, the time required for the droplet to thermally re-equilibrate with the surroundings has been increased due to a reduction in the rate at which heat from the droplet is delivered to the environment that is not compensated for by heat conduction into the low-temperature substrate [6]. On this basis, the temperature is largely retained inside the

droplet, delaying the process of the ice nucleation and growth [7,8].

Moreover, there is certain vapor released during a freezing event for ice to bridge from the micro-nanostructures to the icing droplet, leading to a variation in the wetting behavior of droplets at the interface [6]. The enlargement of wettability increases the solid-liquid contact area, resulting in an augment of heat transfer efficiency. However, the latent heat release induced by nucleation of water molecules at the solid-liquid interface may hinder the growth of ice at interface, resulting in a dry zone between ice and water [9]. This can also reduce the heat transfer efficiency between the droplets and the substrate, effectively delaying the icing process on superhydrophobic surface.

The corresponding statement has been added to the manuscript and supporting file using yellow background.

5. The air flow manipulating surface with microscale steps is significantly smaller than water droplet, so please present the effect of such structure on wettability and mobility of water, is the water droplet having anisotropy in sliding on this surface?

Author reply: Thanks for your suggestions. In order to investigate the motion characteristics of droplets on superhydrophobic surfaces, the German Kruss K100 adhesion tester is adopted to measure the droplet adhesion during the directional movement process. In this work, the movement direction of the droplet facing the windward side of the microstructure is defined as **direction 1**, while the opposite direction is defined as **direction 2** since the arrayed microstructures have directionality. Meanwhile, the directions along the ridge line of the microstructure are defined as

directions 3 and 4, respectively, as shown in Figure 2(a). The droplet used in this test is ultra-pure water with a volume of 20 μL . The droplet movement speed is recognized as 40 mm/min, and the movement distance is set to 18mm to ensure that the droplet always moves within the microstructure region. As a contrast, the dynamic adhesion of droplets in two vertical directions for the superhydrophobic plate with micro-nanostructures is also carried out to verify whether the superhydrophobic micro-nanostructures obtained by electrodeposition have isotropic wettability.

The results indicate that the adhesion forces of droplets moving in two vertical directions on the superhydrophobic plate is 0.0186 N and 0.0193 N, respectively. It is clear that the superhydrophobic micro-nanostructures obtained by electrodeposition have isotropic wettability, as shown in Figure 2(b). However, the adhesion force of the droplet moving along the direction 1 on A-20 sample is 0.0006 N, which is only 13% of that in the opposite direction, as shown in Figure 2(c). This demonstrates that droplets are inclined to slide towards the windward side in the direction of microstructure arrangement. It is worth noting that the droplet adhesion forces are 0.0005 N and 0.00056 N in directions 3 and 4 respectively when droplets move along the ridge line of microstructures, which is little different from the droplet adhesion force in direction 1, as displayed in Figure 2(d). This clarifies that the droplet may move randomly along ridge direction or towards the windward of the microstructures when sliding on the superhydrophobic A-20 surface.

Additionally, the adhesion force of the droplet moving towards the windward side on A-30 sample is 0.0012 N, which is 17.9% of that in the opposite direction, as

illustrated in Figure 2(e). Interestingly, the adhesive force of droplets rolling along the ridge line is between 0.0011 N and 0.0018 N, which is slightly higher than that moving towards the windward side, as demonstrated in Figure 2(f). Hence, unlike the A-20 sample, it can be assumed that the droplet on A-30 sample is apt to sliding towards the windward side. Moreover, the adhesive force is measured to be 0.006 N when the droplet moves towards the windward side on the A-40 sample, while the adhesive force of the droplet rolling in the opposite direction is 0.0099N, as shown in Figure 2(g). Simultaneously, the adhesive force of the droplets moving along the ridge line of the microstructure is only about 0.004 N, which means that the droplets tend to slide along the ridge line of microstructures on the A-40 sample.

Generally, the adhesive force of droplets moving towards the windward side is always smaller than that of droplets rolling in the reverse direction on various superhydrophobic surface with arrayed microstructures. Nevertheless, the adhesive force of droplets shifting along the ridge line of the microstructure is relatively similar. Notably, the difference in adhesion force of droplets moving in directions 1 and 2 gradually decreases and the adhesion force of droplets sliding along the ridge direction also gradually increases with the enlargement of the microstructure angle.

The corresponding statement has been added to the manuscript and supporting file using yellow background.

Figure 2. Dynamic adhesive forces of droplets on different superhydrophobic surfaces: (a) Schematic diagram of the test process, (b) Adhesive forces of droplets

moving in directions 1 and 3 on the superhydrophobic plate, (c) adhesive forces of droplets moving in directions 1 and 2 on the superhydrophobic A-20 sample, (d) adhesive forces of droplets moving in directions 3 and 4 on the superhydrophobic A-20 sample, (e) adhesive forces of droplets moving in directions 1 and 2 on the superhydrophobic A-30 sample, (f) adhesive forces of droplets moving in directions 3 and 4 on the superhydrophobic A-30 sample, (g) adhesive forces of droplets moving in directions 1 and 2 on the superhydrophobic A-40 sample, (h) adhesive forces of droplets moving in directions 3 and 4 on the superhydrophobic A-40 sample

6. During the anti-icing test, the incident angle is too high. In actual application (i.e., aircraft wing), the problem is leading edge, which has 0 degree of incident angle. Thus, more tests with a low incident angle is required to claim anti-icing.

Author reply: Thanks for your suggestions. Considering the directionality of the drag-reduction microstructure, A-20 and A-30 samples are selected for anti-icing experiment in the icing wind tunnel with the incident angles of 0°, 10°, 20°, 30°, 40°, 50°, 60°, 70°, and 80° respectively, in order to investigate the effect of incident angle on the anti-icing performance of the arrayed hierarchical structure, as illustrated in Fig. 3(a). The flow velocity is set as 69.4 m/s and the temperature is defined as -20°C, which is consistent with the previous anti-icing test. The results indicate that the ice accumulation on all samples increases with the decrease of incident angle, as shown in Fig. 3(b)~3(j).

Figure 3. Anti-icing test with various incident angle: (a) Schematic diagram of the test

process, (b) Icing behavior of typical samples at an incident angle of 80° , (c) Icing behavior of typical samples at an incident angle of 70° , (d) Icing behavior of typical samples at an incident angle of 60° , (e) Icing behavior of typical samples at an incident angle of 50° , (f) Icing behavior of typical samples at an incident angle of 40° , (g) Icing behavior of typical samples at an incident angle of 30° , (h) Icing behavior of typical samples at an incident angle of 20° , (i) Icing behavior of typical samples at an incident angle of 10° , (j) Icing behavior of typical samples at an incident angle of 0°

Similar to the previous results, the ice accumulation on the end of the superhydrophobic plate is still higher than that on the front, which is determined by the motion behavior of microdroplets driven by interfacial air flow. Meanwhile, a certain amount of ice accumulation also exhibits on the end of the arrayed hierarchical structure due to the appearance of incident angle. Notably, the ice accumulation on the two arrayed hierarchical structures is lower than that on conventional superhydrophobic surfaces (the ice accumulation reduction is about 30%) at a higher incident angle ($>40^\circ$), demonstrating effective anti-icing applicability, as shown in Fig. 4. With the incident angle further reduces to 40° , the ice accumulation reduction of A-20 sample significant decreases to -29.6%. Subsequent results indicate that the A-20 sample completely loses its superiority in anti-icing when the incidence angle is below 20° . However, it is worth noting that although the A-30 sample shows a slight increase in ice accumulation of 2.62% at an incident angle of 40° , its anti-icing property is again reflected with the further descension of incident angle. Surprisingly, even at an incident angle of 0° , the

ice accumulation on the A-30 sample is reduced by 16.04% compared with that on the conventional superhydrophobic surface, indicating the widespread applicability of the array hierarchical structure in low-temperature and high-velocity inflow environments.

Figure 4. Icing mass reduction of different samples: (a) A-20 sample, (b) A-30 sample

Generally, the interface airflow manipulated by arrayed microstructures can improve the anti-icing performance of conventional superhydrophobic materials in a widespread extent. On this basis, we speculate that arranging the above structures along the wing surface may further reduce the adverse impact of incidence angle on anti-icing performance, as shown in Fig. 5.

Figure 5. Diagram of the arrayed microstructure distributed along the wing surface

The corresponding statement has been added to the manuscript and supporting file using yellow background.

Thanks again for your valuable suggestions and positive evaluation on our research work.

References:

- [1] Yang K, Liu Q, Lin Z, et al. Investigations of interfacial heat transfer and phase change on bioinspired superhydrophobic surface for anti-icing/de-icing. *International Communications in Heat and Mass Transfer*. 134, 105994 (2022).
- [2] Jung S, Tiwari M K, Doan N V, et al. Mechanism of supercooled droplet freezing on surfaces. *Nature communications*. 3, 615 (2012).
- [3] Cohen N, Dotan A, Dodiuk H, et al. Thermomechanical mechanisms of reducing ice adhesion on superhydrophobic surfaces. *Langmuir*. 32, 9664-9675 (2016).
- [4] Boreyko J B, Collier C P. Delayed frost growth on jumping-drop superhydrophobic surfaces. *ACS nano*. 7, 1618-1627 (2013).
- [5] Zhang Y, Yu X, Wu H, et al. Facile fabrication of superhydrophobic nanostructures on aluminum foils with controlled-condensation and delayed-icing effects. *Applied Surface Science*. 258, 8253-8257 (2012).
- [6] Lambley H, Graeber G, Vogt R, et al. Freezing-induced wetting transitions on superhydrophobic surfaces. *Nature Physics*. 1-7 (2023).
- [7] Yin L, Xia Q, Xue J, et al. In situ investigation of ice formation on surfaces with representative wettability. *Applied Surface Science*. 256, 6764-6769 (2010).
- [8] Zhang Q, He M, Zeng X, et al. Condensation mode determines the freezing of condensed water on solid surfaces. *Soft Matter*. 8, 8285-8288 (2012).
- [9] Yang S, Wu C, Zhao G, et al. Condensation frosting and passive anti-frosting. *Cell Reports Physical Science*. 2 (2021).

REVIEWERS' COMMENTS

Reviewer #1 (Remarks to the Author):

All my concerns have been addressed, and I would like to recomned its publication.

Reviewer #2 (Remarks to the Author):

Revised manuscript and response from authors were fully addressed against my comments. Therefore, I believe current version can be published.

REVIEWER COMMENTS

Reviewer #1 (Remarks to the Author):

1. All my concerns have been addressed, and I would like to recommend its publication.

Author reply: Thank you for your recognition! We are grateful for the reviewer's efforts to improve the quality of our paper.

Reviewer #2 (Remarks to the Author):

1. Revised manuscript and response from authors were fully addressed against my comments. Therefore, I believe current version can be published.

Author reply: Many thanks! We are grateful for the reviewer's efforts to improve the quality of our paper.